# TAKE NOTE: YOUR MOLECULAR DATASET IS PROBABLY ALIGNED

**Peter Lippmann, Roman Remme, Manuel V. Klockow & Fred A. Hamprecht**
Interdisciplinary Center for Scientific Computing (IWR), Heidelberg University,
69120 Heidelberg, Germany
`{peter.lippmann,roman.remme,manuel.klockow,fred.hamprecht}`
`@iwr.uni-heidelberg.de`

## ABSTRACT

Massive training datasets are fueling the astounding progress in molecular machine learning. Since these datasets are typically generated with computational chemistry codes which do not randomize pose, the resulting molecular geometries are usually not randomly oriented. While cheminformaticians are well aware of this fact, it can be a real pitfall for machine learners entering the burgeoning field of molecular machine learning. We demonstrate that molecular poses in the popular datasets QM9, QMugs, and OMol25 are indeed biased. While the fact can easily be overlooked by visual inspection alone, we show that a simple classifier can separate original data samples from randomly rotated ones with high accuracy. Second, we empirically validate that neural networks can and do exploit the orientation bias in these datasets by successfully training a model on chemical property prediction using molecular orientation as *sole* input. Third, we present visualizations of all molecular orientations and confirm that chemically similar molecules tend to have similar canonical poses. In summary, we recall and document orientation bias in the prevalent datasets that machine learners should be aware of.

## 1 INTRODUCTION

Machine learning has become a well-established tool for designing, discovering, and studying molecular systems, for instance in drug discovery, materials science and physical chemistry. Much of the progress in the field is enabled by the curation of large-scale datasets that provide accurate molecular properties. Computational chemistry codes used in the underlying data-generating processes usually do not generate molecular geometries in random orientations. At the same time, handling the arbitrariness of coordinate systems by incorporating symmetries into machine learning models has become a central theme in geometric deep learning. SO(3)-equivariant neural networks guarantee well-defined transformation behavior under rotations. In other words, equivariant models produce consistent predictions for inputs that differ only by rotation (or equivalently, by the choice of reference frame). Consequently, equivariant architectures are agnostic to the orientation of molecular geometries in ML datasets. However, most existing equivariant architectures rely on non-standard building blocks, such as specialized normalization layers, nonlinearities, and tensor operations, which are often computationally demanding (Passaro & Zitnick, 2023) and can be challenging to tune in practice (Abramson et al., 2024; Pertigkiozoglou et al., 2024; Elhag et al., 2024). As a consequence, while exact equivariance is desirable in principle, softening the constraint by learning approximate symmetries or breaking built-in equivariance is a (re-)emerging trend in molecular machine learning (Langer et al., 2024; Eissler et al., 2025), fueled by prominent examples such as Alphafold 3 Abramson et al. (2024). Other recent examples include Wang et al. (2023b; 2025); Zhang et al. (2025); Joshi et al. (2025). In that case, possible orientation bias in molecular datasets might affect machine learning workflows. Building on and extending the recent work by Lawrence et al. (2025a), we show that molecules in some of the most popular molecular datasets, including QM9, QMugs and OMol25, are by default not presented in random orientations, and we discuss implications for machine learning practitioners.

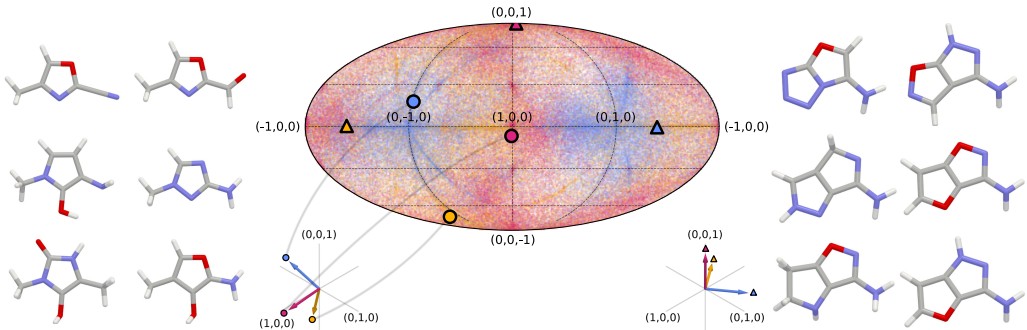

Figure 1: **Molecules in many popular molecular datasets (here: QM9) are not randomly oriented. Structurally similar molecules are oriented similarly.** 2D visualization of normalized principal components (PCs) of all QM9 molecules reveals a clear non-uniform structure. Orientation reference frames correspond to triplets of points in the 2D area-preserving (Mollweide) projection (blue $\sim$ PC1, yellow $\sim$ PC2, magenta $\sim$ PC3). Left and right: two groups of structurally similar molecules that share nearly the same PC orientation (left orientation $\bigcirc$, right orientation $\triangle$).

To date, many architectures for molecular machine learning are benchmarked on the widely used QM9 dataset (Ruddigkeit et al., 2012; Ramakrishnan et al., 2014), a collection of around 134,000 small organic molecules with up to nine heavy atoms. While QM9 is the gold standard for property prediction on smaller molecules, the QMugs collection (Isert et al., 2022) comprises quantum mechanical properties for nearly 2 million larger drug-like molecules with up to 100 heavy atoms extracted from the ChEMBL database (Mendez et al., 2019). The OMol25 dataset (Levine et al., 2025) combines broad chemical diversity with a high level of accuracy at an unprecedented scale (100M systems), comprising not only neutral organics but also biomolecules, metal complexes, and electrolytes, spanning 83 different elements.

We want to raise awareness of the fact that molecules in ML datasets are not oriented randomly for two crucial reasons. First, non-equivariant architectures trained on these datasets must employ explicit data augmentation; otherwise, their test performance will degrade significantly when evaluated on randomly oriented molecules. Second, orientation bias may have an undesired effect on the performance of architectures that introduce symmetry breaking or are only approximately equivariant (Wang et al., 2023b; Pertigkiozoglou et al., 2024; Kaba & Ravanbakhsh, 2024; Elhag et al., 2024; 2025; Lawrence et al., 2025b): for instance, exploiting spurious information contained in biased orientations of molecules may artificially inflate the performance of non-equivariant models. In addition, when molecular properties such as the ground state electron density are evaluated on a grid that is not spherically symmetric (Brockherde et al., 2017; Bogojeski et al., 2020; Jørgensen & Bhowmik, 2022; Li et al., 2025), the canonical orientation may introduce a systematic bias in the grid orientation, even when equivariant architectures are used later in the prediction.

In this paper, we make the following contributions: we demonstrate, using QM9, QMugs, and OMol25 as prominent examples, that molecules in many popular ML datasets are not randomly oriented by training a simple classifier that distinguishes between randomly rotated and unrotated samples with very high accuracy. We show that the accuracy remains high even when the default atom positions are perturbed with substantial noise and random rotations of up to $90°$. Further, we demonstrate that neural networks can leverage the canonical orientation to achieve artificially high accuracy in an extreme scenario: using only the normalized principal components of atom positions as input, we regress molecular properties and observe performance on the three standard datasets that exceeds the best possible accuracy expected for randomly oriented data. Lastly, we visualize the orientations of all molecules in these datasets and show that chemically similar molecules tend to be oriented similarly (see Fig. 1). We make our code publicly available as a toolbox to visualize and quantify orientation bias in molecular datasets at https://github.com/sciai-lab/are-my-molecules-aligned.

## 2 BACKGROUND AND RELATED WORK

Formally, a function $\varphi : V \to W$ between vector spaces $V$ and $W$, is *equivariant* under a group $G$ if $\rho_{\text{out}}(g)\, \varphi(x) = \varphi(\rho_{\text{in}}(g)\, x)$ for all $g \in G$ and $x \in V$. Here, $\rho_{\text{in}}, \rho_{\text{out}}$ are group representations on $V$ and $W$, which define how elements of $G$ act on $V$ and $W$, respectively. In diagrammatic form, equivariance means that the following commutes:

$$
\begin{array}{ccc}
x & \xrightarrow{\ \varphi\ } & \varphi(x) \\
{\scriptstyle \rho_{\text{in}}(g)} \downarrow & & \downarrow {\scriptstyle \rho_{\text{out}}(g)} \\
x' & \xrightarrow{\ \varphi\ } & \varphi(x')
\end{array}
$$

Several approaches to incorporating exact $SO(3)$-equivariance into neural networks exist. Most famously, tensor field networks use tensorial features in all hidden layers to maintain a well-defined transformation behavior throughout the network (Thomas et al., 2018; Geiger & Smidt, 2022; Batatia et al., 2022; Liao et al., 2024; Aykent & Xia, 2025). Similarly, specialized architectures exist to use elements of the projective geometric (or Clifford) algebra to achieve Euclidean rotation and translation equivariance (Brehmer et al., 2023; Ruhe et al., 2023). In contrast, canonicalization approaches avoid the need for specialized architectural building blocks and use canonical reference frames to guarantee exact equivariance (Puny et al., 2021; Pozdnyakov & Ceriotti, 2024; Spinner et al., 2025; Lippmann et al., 2025). Data augmentation offers a simple and practical alternative where equivariance is learned approximately by presenting the network with randomly rotated inputs (and targets). It is an open research question whether built-in equivariance is favorable over data augmentation, and a fair comparison is often non-trivial (Brehmer et al., 2024). Some evidence points to the superiority of non-equivariant architectures (Langer et al., 2024; Lippmann et al., 2025), possibly due to their less constrained design space.

In their recent work, Lawrence et al. (2025a) introduced the idea that orientation biases exist in molecular datasets. While the focus of their analysis lies on the detection of orientation bias, we here introduce a set of alternative and complementary methods to systematically study the distribution of molecular orientations.

Related to broken symmetries in geometric datasets, there are several systems of interest in which symmetries are spontaneously broken, such as dynamical phase transitions (Baek et al., 2017) or polar fluids (Gibb et al., 2024). For these tasks, networks with exact, built-in equivariance might be too constrained and functional symmetry breaking (Wang et al., 2023a) offers a possible solution (Wang et al., 2022). The authors of (Wang et al., 2023a) further introduced the notion of distributional symmetry breaking. The term applies to data distributions under which a data sample and its transformed version are not equally likely, such as particle collisions recorded at particle colliders, where the beam axis singles out a preferred direction (Favaro et al., 2025). Orientation bias in molecular datasets can be seen as another instance of distributional symmetry breaking w.r.t. the symmetry group of rotations.

### 2.1 RELATED WORKS THAT MAY BE AFFECTED BY ORIENTATION BIAS

As pointed out above, one subtle example where an orientation bias may easily be overlooked is charge density prediction. In the field, several standard benchmarks exist that store DFT ground state electron densities on regular Cartesian (axis-aligned) grids as training targets (Brockherde et al., 2017; Bogojeski et al., 2020; Jørgensen & Bhowmik, 2022; Li et al., 2025).

We have investigated all four datasets and found that they are indeed strongly canonicalized (see App. E.1). The combined datasets (Brockherde et al., 2017; Bogojeski et al., 2020) contain molecular dynamics trajectories for six different molecules (around 1000 data samples each). The datasets (Jørgensen & Bhowmik, 2022; Li et al., 2025) both contain electron densities evaluated on regular Cartesian grids for all QM9 molecules. The two datasets differ in the DFT software used to calculate the reference electron density: VASP in (Jørgensen & Bhowmik, 2022) and PySCF in (Li et al., 2025). However, both datasets use the standard QM9 dataset which exhibits orientation bias (cf. Fig. 1). Since the volumetric grids are not spherically symmetric, the evaluation of the loss function is not rotationally-invariant, which may introduce a systematic bias even when equivariant architectures are used later in the prediction. We can think of two scenarios in which such biases may lead to problems: (a) when practitioners deploy trained models, e.g. in MD simulations, they

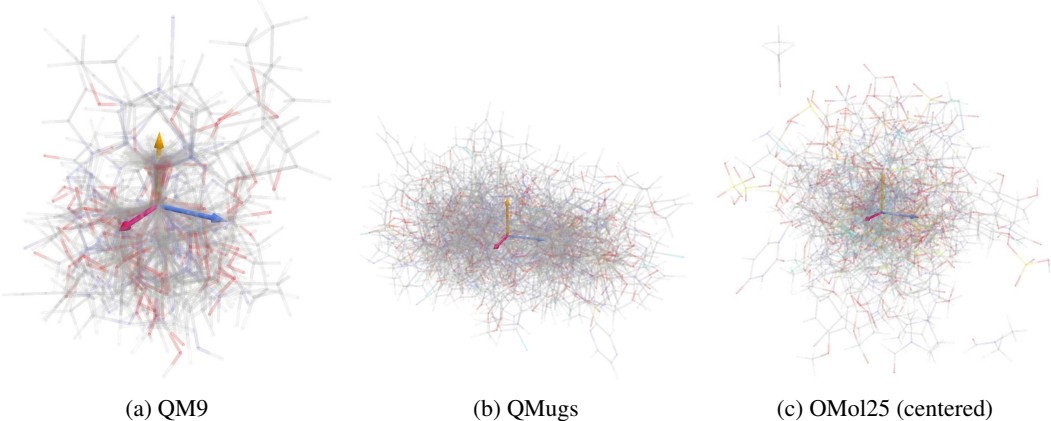

(a) QM9                         (b) QMugs                    (c) OMol25 (centered)

Figure 2: **Overlay of 100 random molecular geometries from QM9, QMugs, and OMol25 respectively.** QM9 shows strong alignment of the bond adjacent to the origin with the Cartesian $y$-axis (yellow). Bulges of QMugs and OMol25 molecules exhibit less structure, but are not spherically symmetric.

will, most likely, not rotate the simulated molecules into a preferred orientation, which may decrease the performance of the model; and (b) when studying extrapolation capabilities of trained models, e.g. for larger molecules, the generalization to new datasets may be hindered by orientation biases.

Furthermore, as mentioned previously, using non-equivariant architectures for (equivariant) molecular tasks has emerged as a recent but growing trend. Some of the proposed architectures do not use explicit data augmentation, making them susceptible to orientation biases in molecular datasets; for instance, the authors of (Pertigkiozoglou et al., 2024) relax the equivariance constraint in the intermediate layers of networks during training by introducing an additional non-equivariant term that is progressively constrained until one arrives at a fully equivariant solution. Therefore, orientation bias in the datasets used (e.g. in MD17, see Fig. 11) could have an undesired effect on the optimization trajectory of such networks. The authors of (Elhag et al., 2024; 2025) present machine learned interatomic potentials (MLIPs) trained to learn rotational equivariance approximately by minimizing an additional loss. The MLIPs are trained on OMol25 and the MD17 dataset, which can both be shown to exhibit strong orientation bias (cf. Fig. 5b and Fig. 11), which may influence the training and evaluation of these models. Motivated by the "bitter lesson" Sutton (2019), the conformer generation model presented in (Wang et al., 2023b) is based on an efficient and scalable diffusion model that operates directly on 3D atomic positions without enforcing rotational equivariance. The authors conduct experiments on QM9 and the strongly aligned GEOM dataset (cf. Fig. 7a). Notably, the authors observe that randomly rotating their training set prior to training negatively impacts their performance and hypothesize that the reason may be that "DFT simulations used to generate the data might be implicitly encoding a canonical coordinate system, which affects generalization if broken" (Wang et al., 2023b) (p. 8).

## 3    INVESTIGATING ORIENTATIONS IN MOLECULAR DATASETS: METHODS, RESULTS, AND IMPLICATIONS

Clearly, the first step that comes to mind when investigating the orientations of molecular geometries is to visually inspect the 3D geometries for obvious alignment. Figure 2 shows 100 randomly sampled molecular geometries from each dataset. For QM9, clear structure is visible. Most strikingly, the first bond (adjacent to the origin) almost perfectly aligns with the Cartesian y-axis. The original QM9 paper (Ramakrishnan et al., 2014) invoked the cheminformatics tool Corina (version 3.491, 2013) (Sadowski & Gasteiger, 1993) to generate 3D structures from SMILES strings. The geometries were then relaxed using Kohn-Sham DFT calculations at the B3LYP/6-31G(2df,p) level. The Corina algorithm (closed-source) is likely responsible for the alignment with the y-axis, while the subsequent geometry relaxation softens the strict alignment. For QMugs and OMol25, no similarly

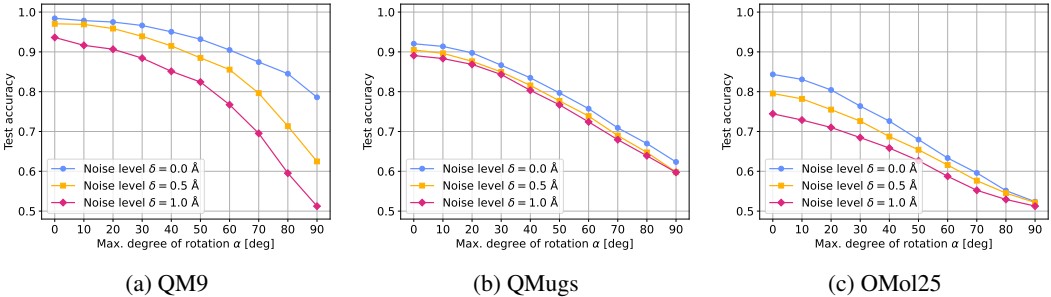

(a) QM9         (b) QMugs         (c) OMol25

Figure 3: **Canonical orientations in QM9, QMugs, and OMol25 are highly consistent and detectable.** A simple geometric message passing network accurately discerns canonical samples from randomly rotated ones, even if the atom positions are previously perturbed and the molecule randomly rotated by an angle up to $\alpha$.

distinct structure is visible regarding the orientation, but the central bulge of molecules is clearly not spherically symmetric, hinting at a systematic orientation bias.

### 3.1 LEARNED DETECTION OF DEFAULT ORIENTATIONS

To empirically validate that molecules in ML datasets exhibit orientation bias, we have trained a simple message passing network to distinguish between molecules in their original ("canonical") orientation and ones that have been randomly rotated. If the dataset were truly orientation invariant, such a classification task would be impossible beyond random guessing. This test is known in the literature as classifier two-sample test (Lopez-Paz & Oquab, 2017) and has previously been applied to molecular datasets by Lawrence et al. (2025a).

For each sample in the dataset[1], we randomly decide whether to apply a global rotation, sampling a rotation matrix uniformly from SO(3). To ensure that the learned detection is not just based on a simple geometric pattern (such as a particular edge being aligned with a coordinate axis, cf. Fig. 2), we apply Gaussian noise (with standard deviation $\delta$ up to 1 Å) to the default atomic positions as well as random rotations up to a maximum angle $\alpha$ (before applying the uniformly sampled rotation that should be detected by the network). A noise level of $\delta = 1$ Å is quite substantial, considering that the length of a carbon-carbon single bond is 1.5 Å, and shorter for a double or triple bond. The network is then trained to minimize a binary cross-entropy loss, predicting whether the input molecule has been randomly rotated or is in its (perturbed) default pose. We employ a straightforward point cloud architecture, consisting of three layers of message passing:

$$f_i^{(k+1)} = \bigoplus_{j \in \mathcal{N}(i)} \mathrm{MLP}(f_j^{(k)}, \mathrm{emb}(x_i - x_j)), \tag{1}$$

where $f_i^{(k)}$ denotes the feature vector of atom $i$ in layer $k$, $x_i$ its position, and $\mathcal{N}(i)$ its neighborhood (defined by a radial cutoff of 10 Å). Importantly, this network is neither rotation-equivariant, nor invariant, but rotation-dependent by design. During message passing, the angular and radial part of the relative distance vectors $x_i - x_j$ are embedded using Gaussian basis functions. Prior to these message passing layers, we combine a learned embedding of each atom's neighborhood geometry with an embedding of the atom type to initialize the node features $f_i^{(0)}$. We use the same architecture for all three datasets, see App. D for details.

The results, summarized in Fig. 3, reveal that even a simple classifier can discern randomly rotated samples from canonical ones with very high test accuracy, even when the default atom positions are substantially perturbed. This provides clear evidence that molecules in the considered datasets are not randomly oriented, and that the canonical poses are highly consistent and detectable.

---

[1] For all our experiments, we use the QM9 dataset readily available in PyTorch Geometric (Fey & Lenssen, 2019), QMugs from `https://doi.org/10.3929/ethz-b-000482129` and OMol25 available through the `fairchem` repository (`https://github.com/facebookresearch/fairchem`).

## 3.2 QUANTITATIVE ORIENTATION ANALYSIS

To systematically study the orientation of molecules in the respective datasets, we devise a mapping $\Omega$ from the set of molecular geometries $\mathcal{M} = \{\{(z_a, x_a)\}_{a \in A}\}$ to the set of orientations $SO(3)$. For each molecule, consisting of atoms $A$ with charges $z_a$ and positions $x_a$, it outputs an orientation. Clearly, for general molecules, no canonical orientation function exists. However, it is very reasonable to require that orientations predicted for two molecules that differ just by a rotation must be consistent. Intuitively, the orientation $\Omega(M)$ can be thought of as a coordinate frame attached to the molecule that should rotate along when the molecule is rotated. Formally, we thus demand that the mapping $\Omega : \mathcal{M} \to SO(3)$ is equivariant under rotations $R$ applied to the molecular geometry:

$$\Omega(RM) := \Omega(\{(z_a, Rx_a)\}_{a \in A}) = \Omega(\{(z_a, x_a)\}_{a \in A})R^{\mathrm{T}} = \Omega(M)R^{\mathrm{T}}. \tag{2}$$

To understand the constraint imposed by Eq. (2), we view $\Omega(M) \in SO(3)$ as a collection of three row vectors $e_i \in \mathbb{R}^3, i = 1, 2, 3$, i.e. $\Omega(M) = (e_1, e_2, e_3)^{\mathrm{T}}$. The $e_i$ are precisely the basis vectors of the reference frame that rotates along with the molecular geometry $M$, i.e.

$$M' = RM \quad \to \quad e'_i = Re_i \quad \text{or equivalently:}$$

$$\Omega(RM) = (Re_1, Re_2, Re_3)^{\mathrm{T}} = (e_1, e_2, e_3)^{\mathrm{T}}R^{\mathrm{T}} = \Omega(M)R^{\mathrm{T}},$$

as demanded per Eq. (2). A simple orientation function is obtained by choosing the basis vectors $e_i$ to be the normalized principal components of the centered atom positions. To account for the fact that the sign of eigenvectors is ambiguous, we choose the sign of the first two principal components such that they point in the direction of the largest absolute projection, that is, we orient them to satisfy

$$\max_{a \in A} |x_a \cdot e_i| = \max_{a \in A} x_a \cdot e_i \quad \text{for } i = 1, 2. \tag{3}$$

The orientation of the third principal component $e_3$ is fixed by the constraint that $\det((e_1, e_2, e_3)^{\mathrm{T}}) = 1$. While one may also use a weighted covariance matrix based on atomic masses (yielding the same eigenvectors as the moment of inertia tensor), we opted for the simpler unweighted version which is robust against exchange of atom types.

To compare the orientations of different molecules, we define a distance measure based on the following fact: Any rotation can be described by a rotation axis (vectors pointing along this axis are left invariant) and the rotation angle, specifying how much to rotate around that axis. This angle is given by $\arccos((\mathrm{tr}(R) - 1)/2)$, see App. A. Then, for two rotations $R_1, R_2 \in SO(3)$, we use the rotation angle of the relative rotation matrix $R_1^T R_2$ as a distance measure:

$$\theta(R_1, R_2) = \arccos\left(\frac{\mathrm{tr}(R_1^{\mathrm{T}} R_2) - 1}{2}\right). \tag{4}$$

Intuitively, $\theta(R_1, R_2)$ is the angle of the rotation that maps from reference frame $R_1$ to $R_2$ and vice versa. The properties of the trace imply that $\theta(R_1, R_2) = \theta(R_2, R_1) = \theta(R_1^{\mathrm{T}}, R_2^{\mathrm{T}})$. If the orientations of molecules in a dataset $\mathcal{D} = \{M_i\}_{i=1}^N$ were truly random, the set of all orientations would be uniformly distributed on $SO(3)$. In that case, for any given reference orientation $\tilde{R}$ the empirical distribution of distances $\{\theta(\tilde{R}, \Omega(M_i)) \mid M_i \in \mathcal{D}\}$ should approximate the following distribution (see App. B):

$$p(\theta) = \frac{2}{\pi} \sin(\theta/2)^2, \quad \theta \in [0, \pi]. \tag{5}$$

To characterize deviations from this distribution, we proceed as follows. For each dataset, we first compute the distance matrix $\Theta_{ij} = \theta(\Omega(M_i), \Omega(M_j))$ for all $M_i, M_j \in \mathcal{D}$ (very large datasets are subsampled randomly). Afterwards, we apply a 1D Gaussian kernel[2] $k(\theta|\mu, \sigma^2)$ centered at $\theta = 0$ to each entry in $\Theta$ and sum over the rows of the resulting matrix to obtain the following kernel density estimate at $\theta = 0$:

$$\mathrm{KDE}(\theta = 0 | M_i) = \frac{1}{N} \sum_{j=1}^N k(0 | \Theta_{ij}, \sigma^2) = \frac{1}{N} \sum_{j=1}^N k(\Theta_{ij} | 0, \sigma^2), \tag{6}$$

$$\text{with kernel } k(\theta | \mu, \sigma^2) = \frac{1}{\sqrt{2\pi\sigma^2}} \exp\left(-\frac{(\theta - \mu)^2}{2\sigma^2}\right). \tag{7}$$

---

[2]Given that we work with small $\sigma$, we approximate the von Mises distribution on the circle with a Gaussian kernel.

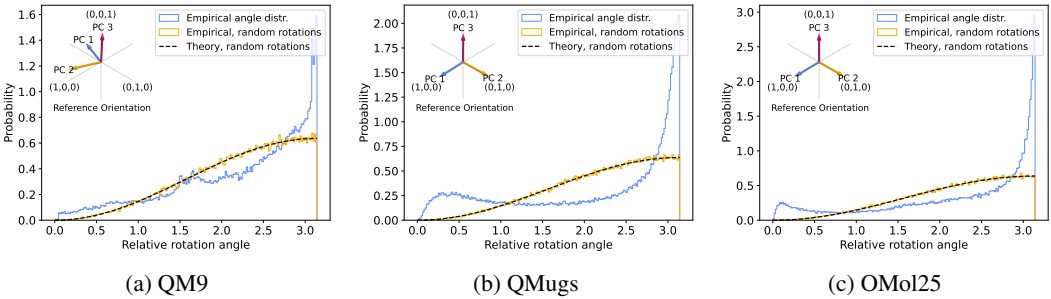

(a) QM9  (b) QMugs  (c) OMol25

Figure 4: **Quantitative comparison of canonical orientations vs. uniform random orientations.** The empirical distribution of relative angles between each orientation in the dataset and the most common reference orientation differs significantly from the same distribution for a randomly rotated dataset and from the theoretical expectation for uniform random orientations. Histograms for QMugs and OMol25 are based on 130,000 randomly sampled molecules.

The most prominent orientation in the dataset is then given by

$$\Omega(M_{i^*}) \text{ with } i^* = \arg\max_{i \in \mathcal{D}} \mathrm{KDE}(\theta = 0 | M_i). \tag{8}$$

Empirically, we find that $\Omega(M_{i^*})$ is fairly robust against the choice of $\sigma$. Using $\Omega(M_{i^*})$ as reference rotation, the distribution of relative rotation angles in row $\Theta_{i^*}$ differs strongly from the theoretically expected distribution for uniformly sampled orientations given by Eq. (5), see Fig. 4. For both QMugs and OMol25, the most common principal component directions (using $\sigma = 0.5$) indeed align closely with the standard Cartesian coordinate frame ($e_1 = (1, 0, 0)^{\mathrm{T}}$, $e_2 = (0, 1, 0)^{\mathrm{T}}$, $e_3 = (0, 0, 1)^{\mathrm{T}}$), see also Fig. 5. For QM9, the most common principal component direction is more ambiguous (compare Figs. 1 and 5). Our analysis has identified the most common principal component orientation in QM9 to be approximately $e_1 = (-0.34, -0.94, 0)^{\mathrm{T}}$, $e_2 = (0.94, -0.34, 0)^{\mathrm{T}}$, $e_3 = (0, 0, 1)^{\mathrm{T}}$. All three datasets contain significantly more molecules in the most common orientation than expected for a uniform distribution of orientations as well as significantly more orientations that differ by a rotation of $180°$, forming the peak at $\theta = \pi$. The latter likely correspond to the same principal components, but with orientations flipped relative to the reference.

**Measuring the non-uniformity of orientations in a single number.** To quantify the deviation of the empirical distribution of orientations (described by rotation matrices) from the uniform distribution (Haar measure) on $\mathrm{SO}(3)$, we estimate the Kullback-Leibler (KL) divergence between the empirical sample distribution and the uniform reference distribution. As an empirical estimator, we employ a version of the classical Kozachenko-Leonenko estimator (Kozachenko, 1987) adapted to the curved domain of $\mathrm{SO}(3)$ (see App. C.1 for details). Our estimated KL divergences align very well with the visual impressions from the Mollweide plots. We obtain estimates of 0.90 for the QM9 dataset, 1.76 for QMugs (from 130,000 samples), and 1.04 for OMol25 (from 130,000 samples). A higher number indicates a stronger non-uniformity of orientations in the dataset. Furthermore, sorting the OMol25 subsets by this non-uniformity measure yields a visually convincing ordering (see Fig. 7). In addition, we have devised a Kolmogorov–Smirnov test to compare the empirical distance distribution between orientations and the expected reference given by Eq. (5) and have studied the robustness of both (non-)uniformity tests (see App. C.2 and C.3). Alternative statistical tests for quantifying distributional symmetry are based on kernel methods, such as maximum mean discrepancy (MMD), that base their estimates on empirical means over distances between individual samples Chiu & Bloem-Reddy (2023); Soleymani et al. (2025); Lawrence et al. (2025a).

### 3.3 EXPLOITING THE CANONICAL ORIENTATION FOR PROPERTY PREDICTION

While our previous investigations demonstrate that molecules in QM9, QMugs, and OMol25 are not randomly oriented, they do not yet address the possible impact on the performance of machine learning models. In the following, we will investigate whether neural networks can exploit the "canonical" orientation of molecules in typical machine learning tasks such as molecular property prediction.

Table 1: **Molecular property prediction from a molecule's orientation alone.** A simple MLP trained on the canonical datasets to regress molecular properties using *only* the normalized principal components of atomic positions as input significantly outperforms the test performance achievable with uninformative input features (mean $\pm$ std over 5 runs).

| Dataset | Property | Random orientation | MSE of mean (test) | MSE of MLP (test) |
|---|---|---|---|---|
| QM9 | $\epsilon_{\text{LUMO}}$ [eV] | no | 1.6355 | **1.4237 $\pm$ 0.0048** |
| | | yes | 1.6355 | 1.6367 $\pm$ 0.0001 |
| QM9 | ZPVE [eV] | no | 0.8107 | **0.6204 $\pm$ 0.0011** |
| | | yes | 0.8107 | 0.8111 $\pm$ 0.0001 |
| QM9 | $c_V$ $\left[\frac{\text{cal}}{\text{mol K}}\right]$ | no | 16.169 | **13.814 $\pm$ 0.083** |
| | | yes | 16.169 | 16.173 $\pm$ 0.001 |
| QMugs | $U_{RT}$ [$E_h$] | no | 890.54 | **843.48 $\pm$ 0.09** |
| | | yes | 890.54 | 890.55 $\pm$ 0.02 |
| QMugs | $\hat{V}_{ee}$ [$E_h$] | no | $4{,}894.0 \times 10^3$ | **$(4{,}596.6 \pm 0.7) \times 10^3$** |
| | | yes | $4{,}894.0 \times 10^3$ | $(4{,}894.6 \pm 0.2) \times 10^3$ |
| OMol25 | $E_{\text{tot}}$ [eV] | no | $14{,}394.3 \times 10^6$ | **$(13{,}689.1 \pm 1.7) \times 10^6$** |
| | | yes | $14{,}394.3 \times 10^6$ | $(14{,}398.7 \pm 0.3) \times 10^6$ |

Here, we consider an extreme scenario: Can a network learn anything about molecular properties when presented *with the orientation of the molecule alone*, without any information about the molecule's constitution or geometry? In the following, we investigate the regression performance of a model that receives *only* normalized principal components of the atom positions as input features (cf. Sec. 3.2), once in canonical pose and once after random rotation. For a fair comparison, we use a deterministic rotation conditioned on the molecule index so that the same sample will be transformed with the same rotation when revisited during training. This is equivalent to using a version of the dataset in which each molecule has been rotated once prior to training. For molecules in random orientations, the normalized principal components do not contain any chemically relevant information. Under an MSE (mean squared error) loss, the best a model can do in this setting is to predict the mean of the target feature, since

$$\text{mean}(\{y_i\}) = \arg\min_x \sum_i (x - y_i)^2. \tag{9}$$

Therefore, if the trained model achieves a significantly better MSE on the test set than the mean of the test targets does, the model has learned a non-trivial pattern from the normalized principal components. This indicates that chemically similar molecules, by default, tend to have similar orientations. The results, presented in Tab. 1, reveal precisely that. Indeed, simple MLPs trained and tested on the canonical datasets significantly outperform the theoretically best possible results, while models trained on the transformed datasets do not, as expected. The chemical properties used for regression were chosen based on the amount of structure in the visualization of all molecular orientations when using the respective properties as a heat map (see Sec. 3.4).

This provides empirical evidence that neural networks may learn an unphysical pattern by mapping canonical orientations to chemical properties. In that case, the spurious information contained in the biased orientations could be exploited by non-equivariant models, potentially leading to artificially inflated performance metrics in the absence of a randomly oriented test set. In an additional experiment, we have studied to what extent a (non-equivariant) transformer architecture trained on property prediction can be influenced by the orientation bias in the selected datasets (see App. E.5 for details).

## 3.4 VISUALIZATION OF MOLECULAR ORIENTATIONS

To further investigate whether chemically similar molecules in molecular datasets tend to be similarly oriented, we have devised the following visualization strategy: for every molecule $M$, we

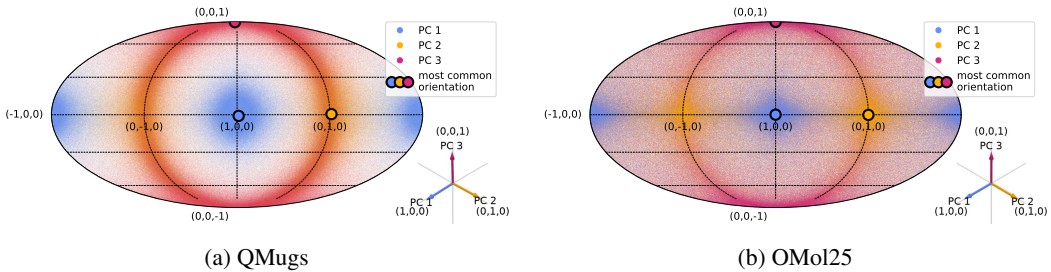

(a) QMugs       (b) OMol25

Figure 5: **2D visualization of normalized principal components of all molecules in QMugs and $10^6$ random samples from OMol25.** The equal-area Mollweide projection of the three principal axes reveals that the most common principal axes orientation aligns with the standard Cartesian coordinate system. Orientation frames consist of a triplet of points in the 2D projection (blue $\sim$ PC1, yellow $\sim$ PC2, magenta $\sim$ PC3).

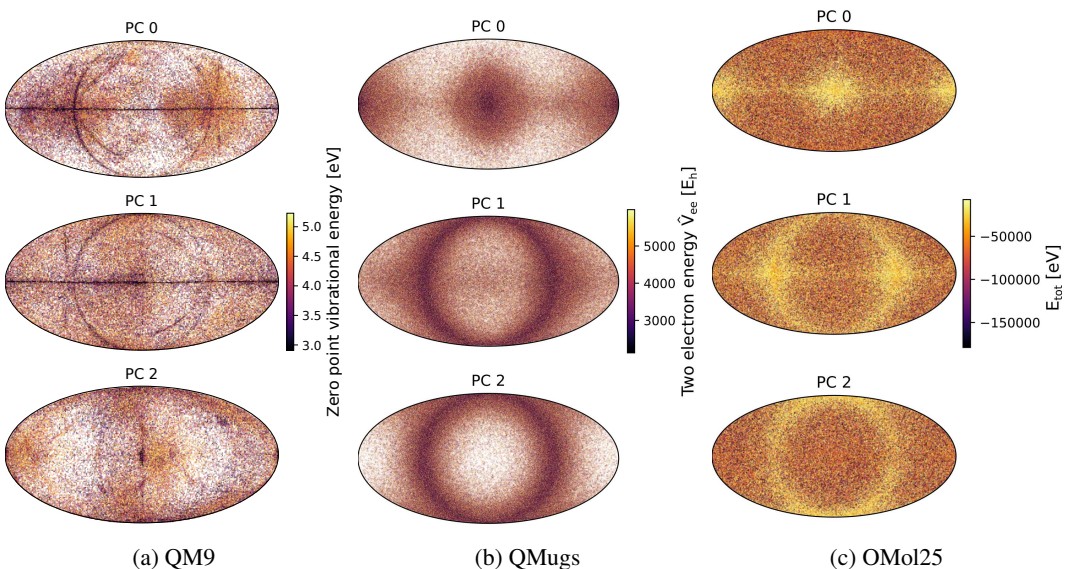

(a) QM9       (b) QMugs       (c) OMol25

Figure 6: **Default orientations of molecules are correlated with chemical properties.** Plots are colored by chemical properties in in the 2D Mollweide projections of the three normalized principal component directions to reveal substructure within the distributions. Non-equivariant architectures may exploit such correlation to artificially inflate performance.

compute the normalized principal components $e_1, e_2, e_3 \in \mathbb{R}^3$ and combine them into a rotation matrix $\Omega(M) = (e_1, e_2, e_3)^{\mathrm{T}}$, as described in Sec. 3.2. The $e_1, e_2, e_3 \in \mathbb{S}^2$ are projected using the equal-area Mollweide projection, and are each visualized in a different color (blue $\sim$ PC1, yellow $\sim$ PC2, magenta $\sim$ PC3), such that an orientation frame $\Omega(M)$ consists of a triplet of orthogonal points in the projection, see Figs. 1 and 5. Notably, using an equal-area projection, a perfectly uniform distribution would also be perceived as a uniform distribution. In contrast, the visualizations show a clear pattern and illustrate the previous finding (Sec. 3.2) that the most common principal component orientation in the QMugs and OMol25 datasets aligns with the standard Cartesian coordinate system. Similarly, Fig. 1 reveals the structure in all principal component orientations for QM9. Based on the distance measure defined in Eq. (4), we have cherry-picked two groups of QM9 molecules of practically identical orientation that arguably have very similar chemical constitution and geometry. These examples nicely illustrate that, as one of the signatures of the cheminformatics codes used in the data-generating process, chemically similar molecules tend to have similar canonical orientations. In Fig. 6 we show one selected chemical property for each dataset as a heat map in the projections instead of using distinct colors to distinguish the principal components. The three

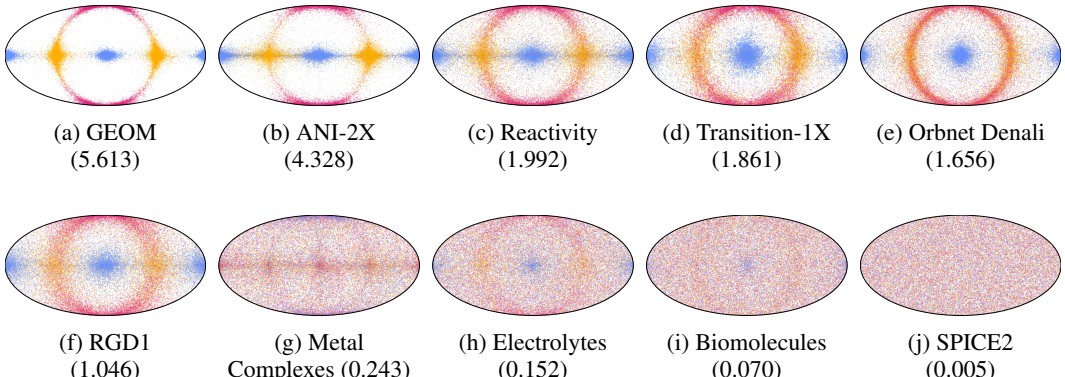

(a) GEOM
(5.613)

(b) ANI-2X
(4.328)

(c) Reactivity
(1.992)

(d) Transition-1X
(1.861)

(e) Orbnet Denali
(1.656)

(f) RGD1
(1.046)

(g) Metal
Complexes (0.243)

(h) Electrolytes
(0.152)

(i) Biomolecules
(0.070)

(j) SPICE2
(0.005)

Figure 7: **Visualization of molecular orientations in OMol25 subsets.** Molecules in some subsets display a strong alignment of principal components (PCs) with the standard Cartesian coordinate system (a, b, c, d, e, f). For others, orientations are more uniformly distributed (g, h, i, j), as indicated by the estimated KL divergence to the uniform distribution over rotations (in brackets, see App. C.1). PCs are projected using the equal-area Mollweide projection and colored as in Fig. 5.

visualizations visibly confirm the correlation between chemical properties of molecules and their default orientations.

Lastly, we show that the orientation distribution in the large collection of OMol25 differs strongly between different subsets of the dataset (subsets are based on the "data_id" field of OMol25 samples), see Fig. 7. The plots demonstrate that while molecules in some subsets are visibly strongly aligned with the standard Cartesian axes, for other subsets the distribution of orientation is (almost) perceptually uniform (e.g. for the SPICE2 dataset (Eastman et al., 2023), Fig. 7(j)). In particular small biases as in the Biomolecules subset (Fig. 7(i)) may be easily overlooked, highlighting the need for random rotations even in the absence of an obvious alignment.

## 4    CONCLUSION

We demonstrate in various ways that the default orientations of molecules in some of the most popular molecular datasets (QM9, QMugs, OMol25) are far from random, and that the alignment of chemically similar molecules can in principle be exploited by machine learning models. Extending the work by Lawrence et al. (2025a), this paper presents a systematic analysis of orientations in the selected datasets. We introduce an interpretable visualization method to jointly plot all molecular orientations. We summarize the distribution of orientations by pairwise distances to identify the most common poses and quantify the degree of orientation bias. In addition, we highlight different scenarios in which ML pipelines may be affected by orientation bias. Given that universally agreed-upon canonical orientations do not—and probably cannot—exist, the presence of orientation bias is not a deficit of these important community resources. However, for researchers entering the field of molecular machine learning, the assumption that molecular poses are fully random can be a significant source of error. Based on our findings, we thus recommend the following best practices for rigorous evaluation and development of molecular machine learning models: First, it is essential to report equivariance errors for equivariant models, and as a sanity check, evaluate equivariant models on randomly oriented test sets. This ensures that any claimed equivariant behavior is genuine and bug-free. Second, for non-equivariant or only approximately equivariant models, it is crucial to use data augmentation during training to prevent overfitting to any canonical orientations and to provide a more realistic assessment of model generalization. It is quite likely that other data-generating processes may also introduce preferred orientations. Therefore, when in doubt, we recommend following the same best practices for other geometric datasets as well.

At the same time, our results highlight the potential benefits of leveraging a well-defined canonical orientation, as explored in recent work (e.g. by Baker et al. (2024)). In scenarios where a meaningful canonicalization is available and justified by the application, it can be advantageous to incorporate this information explicitly.

ACKNOWLEDGMENTS

This study has received funding from the Klaus Tschira Stiftung gGmbH (Simplaix project) and the Wildcard program from the Carl Zeiss Stiftung. Further, this work is supported by Deutsche Forschungsgemeinschaft (DFG) under Germany's Excellence Strategy EXC-2181/1 – 390900948 (the Heidelberg STRUCTURES Excellence Cluster) as well as under project number 240245660 – SFB 1129. The authors acknowledge support by the state of Baden-Württemberg through bwHPC and the German Research Foundation (DFG) through grant INST 35/1597-1 FUGG.

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

## A    DERIVATION OF THE ROTATION ANGLE OF A GIVEN ROTATION MATRIX

Let $R \in \mathrm{SO}(3)$ be a rotation matrix. Since $R$ is real orthogonal, all its eigenvalues lie on the unit circle and complex ones occur in conjugate pairs. With $\det R = 1$, the three eigenvalues must be $\{1, e^{i\varphi}, e^{-i\varphi}\}$ for some $\varphi \in [0, \pi]$. Let $u$ be a unit eigenvector with $Ru = u$ (the rotation axis). Now, let us extend $u$ to an orthonormal basis $\{e_1, e_2, u\}$ with appropriate basis vectors $e_1$ and $e_2$. In this basis $R$ leaves $\mathrm{span}\{e_1, e_2\}$ invariant and acts on the corresponding subspace as a $2 \times 2$ planar rotation by $\theta$. Hence $R$ is (by change of basis) orthogonally similar to

$$\begin{pmatrix} \cos\theta & -\sin\theta & 0 \\ \sin\theta & \cos\theta & 0 \\ 0 & 0 & 1 \end{pmatrix}.$$

Since the trace is invariant under similarity, we have

$$\mathrm{tr}(R) = \cos\theta + \cos\theta + 1 = 1 + 2\cos\theta, \tag{10}$$

which yields the rotation angle of $R$

$$\theta = \arccos\left(\frac{\mathrm{tr}(R) - 1}{2}\right) \in [0, \pi]. \tag{11}$$

Furthermore, the fact that the trace of $R$ is also given by the sum of its eigenvalues $\mathrm{tr}(R) = 1 + e^{i\varphi} + e^{-i\varphi} = 1 + 2\cos(\varphi)$ reveals that $\varphi = \theta$.

## B    DERIVATION OF ANGLE DISTRIBUTION FOR UNIFORMLY SAMPLED ROTATIONS

Let $R \in \mathrm{SO}(3)$ be Haar–uniform, i.e. drawn from the unique probability measure on the rotation group that is invariant under multiplying by any fixed rotation on the left or right. Further, we identify each $R \in \mathrm{SO}(3)$ with a unit quaternion $q = (w, \vec{v}) \in \mathbb{S}^3 \subset \mathbb{R}^4$ modulo the antipodal map $q \sim -q$. The rotation angle $\theta \in [0, \pi]$ of $R$ is related to $q$ by

$$\theta = 2\arccos(|w|). \tag{12}$$

Further, the rotation axis $\vec{n}$ of $R$ is related to $\vec{v}$ by $\vec{n} = \vec{v}/\|\vec{v}\|_2$. Now, if we parametrize the 3-sphere by $q = (\cos\chi, \sin\chi\,\vec{n})$ with $\vec{n} \in \mathbb{S}^2$ and $\chi \in [0, \pi]$ the polar angle as measured from the "north pole" in the $w$-direction, the uniform surface element on $\mathbb{S}^3$ factorizes as

$$d\sigma_{\mathbb{S}^3} = \sin^2\chi\,d\chi\,d\Omega_2, \tag{13}$$

where $d\Omega_2$ is the uniform measure on $\mathbb{S}^2$. Since $w$ is directly related to the rotation angle $\theta$ by Eq. (12), we are interested in the marginal distribution of $w$. To get the marginal of $w$, we compute the area (hence the probability mass for the uniform measure) of the "spherical band" between $\chi$ and $\chi + d\chi$. This area is proportional to

$$\sin^2\chi\,d\chi = \sin^2\chi\left|\frac{d\chi}{dw}\right|dw = \sin^2\chi\,\frac{1}{\sin\chi}\,dw = \sin\chi\,dw = \sqrt{1 - w^2}\,dw, \tag{14}$$

where we have used that $w = \cos\chi$ and that $dw = -\sin\chi\,d\chi$. The marginal density of $w$ is given by

$$f_w(w) \propto \sqrt{1 - w^2}, \qquad w \in [-1, 1]. \tag{15}$$

Normalizing with $\int_{-1}^{1}\sqrt{1 - w^2}\,dw = \pi/2$ gives

$$f_w(w) = \frac{2}{\pi}\sqrt{1 - w^2}, \qquad w \in [-1, 1]. \tag{16}$$

Now, since $q$ and $-q$ represent the same rotation, let us consider the density of $|w|$:

$$f_{|w|}(u) = 2f_w(u) = \frac{4}{\pi}\sqrt{1 - u^2}, \qquad u \in [0, 1]. \tag{17}$$

Using that $u = |w| = \cos(\theta/2)$ (cf. Eq. (12)) and thus $\mathrm{d}u/\mathrm{d}\theta = -\frac{1}{2}\sin(\theta/2)$, the density of $\theta \in [0, \pi]$ follows by change of variables:

$$
\begin{aligned}
p(\theta) &= f_{|w|}\big(\cos(\theta/2)\big) \left| \frac{\mathrm{d}}{\mathrm{d}\theta} \cos(\theta/2) \right| \\
&= \frac{4}{\pi} \sqrt{1 - \cos^2(\theta/2)} \cdot \frac{1}{2} \sin(\theta/2) \\
&= \frac{2}{\pi} \sin^2\big(\theta/2\big), \qquad \theta \in [0, \pi].
\end{aligned}
\tag{18}
$$

Hence, the principal rotation angle of a Haar–uniform $R \in \mathrm{SO}(3)$ has density $p(\theta) = \frac{2}{\pi}\sin^2(\theta/2)$ on $[0, \pi]$.

## C  MEASURING THE NON-UNIFORMITY OF ORIENTATIONS IN A SINGLE NUMBER

### C.1  EMPIRICAL ESTIMATE OF THE KL DIVERGENCE TO THE UNIFORM DISTRIBUTION OF ROTATIONS

To quantify the deviation of the empirical distribution of orientations (described by rotation matrices) from the uniform distribution (Haar measure) on $\mathrm{SO}(3)$, we estimate the Kullback-Leibler (KL) divergence between the empirical sample distribution, denoted by $P$, and the uniform reference distribution $U$.

Given that $\mathrm{SO}(3)$ is a compact manifold with finite volume, the KL divergence can be expressed solely in terms of the differential entropy $H(P)$:

$$
D_{KL}(P \| U) = \log(8\pi^2) - H(P),
\tag{19}
$$

where $8\pi^2$ is the total volume of $\mathrm{SO}(3)$ under the standard bi-invariant metric, and $H(P)$ is the geometric differential entropy. A KL divergence of $0$ indicates perfect uniformity, while higher values indicate clustering or anisotropy in the rotational distribution.

To estimate $H(P)$ from a finite set of samples $\{R_i\}_{i=1}^N$, we employ a non-parametric $k$-Nearest Neighbor ($k$-NN) estimator. We follow the manifold-corrected approach described by Singh & Póczos (2016); Heinz & Grubmüller (2019), which extends the classical Kozachenko-Leonenko estimator (Kozachenko, 1987) to the curved geometry of the rotation group. Unlike standard Euclidean estimators for $\mathbb{R}^d$, which assume that local volumes scale as $r^d$, this method explicitly corrects for the manifold's curvature by using the exact volume of a geodesic ball on $\mathrm{SO}(3)$. The estimator is defined as:

$$
\hat{H}(P) = \psi(N) - \psi(k) + \frac{1}{N} \sum_{i=1}^N \log\left(V(\theta_{ik})\right),
\tag{20}
$$

where $\psi$ is the digamma function (defined as the logarithmic derivative of the gamma function). Further, $N$ is the number of samples, $k$ is the number of neighbors (set to $k = 5$ for all experiments), $\theta_{ik}$ is the geodesic distance from sample $R_i$ to its $k$-th nearest neighbor and $V(\theta_{ik})$ is the volume of a geodesic ball with radius $\theta_{ik}$.

The geodesic distance $\theta_{ij}$ between two rotation matrices $R_i$ and $R_j$ is defined as the angle of the relative rotation (as in Eq. (4))

$$
\theta_{ij} := \theta(R_i, R_j) = \arccos\left(\frac{\mathrm{tr}(R_i^{\mathrm{T}} R_j) - 1}{2}\right), \quad \theta_{ij} \in [0, \pi].
\tag{21}
$$

**Geodesic ball volume on SO(3).** Crucially, the volume $V(\theta)$ of a geodesic ball with radius $\theta$ on $\mathrm{SO}(3)$ deviates from the Euclidean approximation ($\frac{4}{3}\pi\theta^3$) due to the positive curvature of the manifold. In the following, we sketch how the exact volume formula is derived from the integration of the Haar measure. The Haar measure on a compact Lie group provides an intrinsic notion of "volume": it assigns the same measure to any set $S \subset \mathrm{SO}(3)$ and its left- (or right-)translated copy, i.e. $gS = \{gs \mid s \in S\}$ or $Sg = \{sg \mid s \in S\}$ for a given $g \in \mathrm{SO}(3)$. Intuitively, this is the

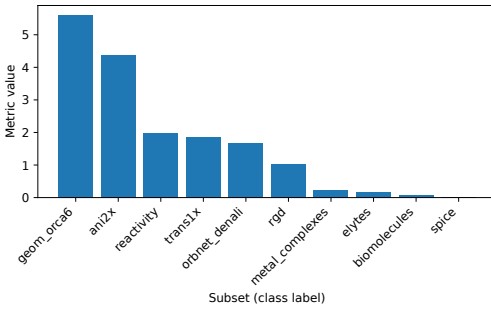
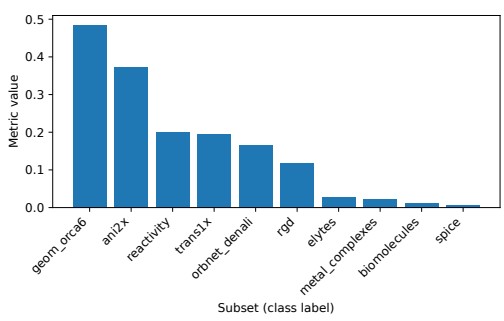

(a) Kullback-Leibler divergence      (b) Kolmogorov–Smirnov test statistic

Figure 8: **Comparison of estimated non-uniformity of orientation distributions of OMol25 subsets.** Estimates (KL divergence and KS test statistic) are based on 19,000 samples from each subset. The estimates imply an ordering that is compatible with the visual impressions of the distributions in Fig. 7.

unique way of measuring subsets of $SO(3)$ that does not depend on the chosen coordinate system or reference frame.

Let $\mu$ denote the (unnormalized) Haar volume measure on $SO(3)$ and let $Vol(SO(3)) := \mu(SO(3))$. We now fix a reference rotation $\tilde{R}$ (e.g. the identity) and define the geodesic ball

$$B_\theta(\tilde{R}) := \{R \in SO(3) : d(\tilde{R}, R) \leq \theta\},$$
$$V(\theta) := \mu(B_\theta(\tilde{R})), \qquad \theta \in [0, \pi].$$

If $R$ is Haar-uniformly distributed, then the probability that $d(\tilde{R}, R) \leq \theta$ is given by

$$\mathbb{P}\big(d(\tilde{R}, R) \leq \theta\big) = \frac{V(\theta)}{Vol(SO(3))}.$$

Hence $V(\theta)$ is proportional to the CDF of the random distance $d(\tilde{R}, R)$:

$$V(\theta) = Vol(SO(3)) \int_0^\theta p(t)\, dt, \tag{22}$$

where $p$ is the distance (angle) density derived in App. B:

$$p(\theta) = \frac{2}{\pi} \sin^2(\theta/2), \qquad \theta \in [0, \pi].$$

The integral can be easily computed as

$$\int_0^\theta p(t)\, dt = \frac{1}{\pi}\big(\theta - \sin\theta\big).$$

Finally, using the known result that the total Haar volume of $SO(3)$ is $Vol(SO(3)) = 8\pi^2$ (see e.g. Eq. (17) in Ecker & Kolev (2025)), we obtain the exact geodesic ball volume

$$V(\theta) = 8\pi\big(\theta - \sin\theta\big), \qquad \theta \in [0, \pi]. \tag{23}$$

As a sanity check, for small $\theta$ we indeed recover the Euclidean approximation:

$$V(\theta) = 8\pi\big(\theta - \sin\theta\big) = 8\pi\Big(\theta - \big(\theta - \tfrac{\theta^3}{6} + \mathcal{O}(\theta^5)\big)\Big) = \frac{4}{3}\pi\theta^3 + \mathcal{O}(\theta^5). \tag{24}$$

Substituting Eq. (23) into Eq. (20) yields a statistically consistent, coordinate-free estimate of the entropy.

## C.2 KOLMOGOROV–SMIRNOV TEST FOR NON-UNIFORMITY OF ORIENTATIONS

In Sec. 3.2, we describe how to characterize the distribution of all orientations in a molecular dataset in terms of all pairwise distances between the associated rotation matrices. The quadratic distance matrix $\Theta \in \mathbb{R}^{N \times N}$ (for $N$ orientations) holds the geodesic distance between any two orientations $R_i, R_j \in \mathrm{SO}(3)$ given by the rotation angle of the relative rotation (cf. Eq. (4)):

$$\Theta_{ij} = \theta(R_i, R_j) = \arccos\left(\frac{\mathrm{tr}(R_i^{\mathrm{T}} R_j) - 1}{2}\right). \tag{25}$$

Each row (or column) of $\Theta_{ij}$ may be interpreted as samples from the empirical marginal "radial" distribution of the full distribution of orientations relative to the respective reference orientation.

Furthermore, we show in App. B that, given a sample from the Haar-uniform distribution on $\mathrm{SO}(3)$, the distribution of distances relative to *any* reference orientation should approximate

$$p(\theta) = \frac{2}{\pi} \sin(\theta/2)^2, \quad \theta \in [0, \pi]. \tag{26}$$

We can therefore compare each empirical distribution of distances (given by a row of $\Theta_{ij}$) to the expected marginal of the uniform distribution to quantify the deviation from $\mathrm{SO}(3)$-uniformity. Since the radial distance distributions are 1D, it is straightforward to compute their empirical cumulative distribution functions (CDFs) and compare them in a (one-sample) Kolmogorov–Smirnov (KS) test against the known theoretical reference given by

$$F(\theta) = \int_0^\theta \frac{2}{\pi} \sin(\theta'/2)^2 \mathrm{d}\theta' = \frac{1}{\pi}(\theta - \sin(\theta)), \quad \theta \in [0, \pi]. \tag{27}$$

The empirical CDF $\tilde{F}_i$ of the $i$-th row of $\Theta$ is given by

$$\tilde{F}_i(\theta) = \frac{\text{number of entries in } \Theta_{i,:} \text{ with } \Theta_{ij} \leq \theta}{N} = \frac{1}{N} \sum_{j=1}^N \mathbb{1}_{[0,\theta]}(\Theta_{ij}), \tag{28}$$

where $\mathbb{1}_{[0,\theta]}(\Theta_{ij})$ is the indicator function, equal to 1 if $\Theta_{ij} \leq \theta$ and equal to 0 otherwise.

The Kolmogorov–Smirnov test statistic is then given by

$$D_i = \sup_\theta |\tilde{F}_i(\theta) - F(\theta)|. \tag{29}$$

In Sec. 3.2 we describe how to obtain an estimate for the most common pose in a dataset using a kernel density estimate. We plot the distance distribution relative to the most common orientation and compare it against the reference distribution (Fig. 4). Here, we want to leverage the $D_i$ from all empirical CDFs to obtain an estimate that is not sensitive to a selected reference orientation. A natural way to combine all $D_i$ is the average:

$$D = \frac{1}{N} \sum_{i=1}^N D_i. \tag{30}$$

Higher values for $D$ indicate a larger discrepancy from the uniform distribution. We have applied this KS test and the KL divergence estimation to the subsets of the OMol25 dataset (see Fig. 8). The ranking obtained from both methods is consistent, except for one transposition in the ordering of the "Metal complexes" and the "Electrolytes" subset, whose relative ordering is arguably also difficult to judge visually (Fig. 7g, h).

For the KS test, it is standard procedure to determine the significance of the test in terms of p-values, e.g. from `scipy.stats.kstest`. In our case, the null hypothesis is that a sample is distributed according to the reference distribution, and hence for p-values below the threshold of $0.05$ the null hypothesis can be rejected, and we can say with statistical significance that the sample is not uniformly distributed. We have tested the significance of the non-uniformity in the OMol25 subsets in the following way: first, we split each subset (we use 19,000 samples) in two halves, (a) and (b). With split (a) we determine an independent estimate of the most common orientation

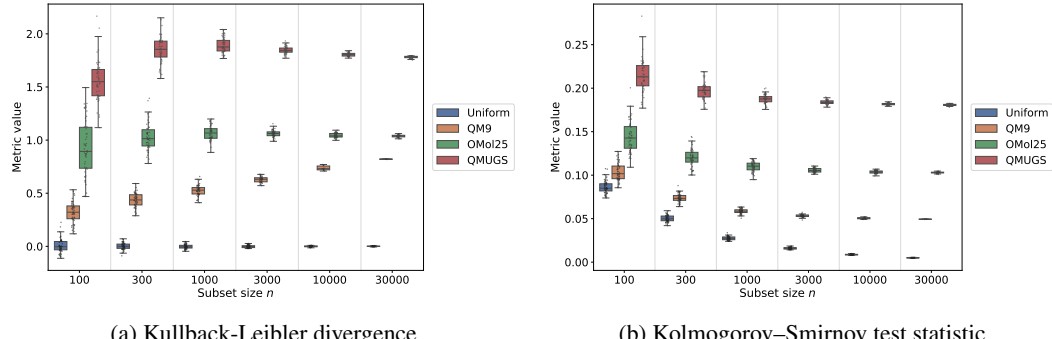

(a) Kullback-Leibler divergence        (b) Kolmogorov–Smirnov test statistic

Figure 9: Robustness of KL divergence and Kolmogorov–Smirnov (KS) test as measure of (non-)uniformity for the orientations in selected molecular datasets. Higher values indicate a larger discrepancy to the Haar-uniform distribution on $SO(3)$.

in the subset (as described in Sec. 3.2). Second, we compare the distribution of distances in split (b) relative to the determined reference orientation via a single (one-sample) KS test and determine the p-value. The independence of the reference orientation is important to not artificially bias the KS-test towards a non-uniform distribution. The p-values and KS test statistics for each subset are shown in Tab. 2.

In App. C.3, we test the robustness of the Kolmogorov–Smirnov test presented here and the empirical estimate of the KL divergence presented in App. C.1 across different sample sizes.

Table 2: **Significance of non-uniformity in OMol25 subsets.** Canonical orientations in all but the SPICE subset differ significantly from the uniform distribution of rotations.

| Data Subset | KS Statistic | p-value |
|---|---|---|
| geom_orca6 | 0.5325 | 0.0000e+00 |
| ani2x | 0.4106 | 0.0000e+00 |
| reactivity | 0.3010 | 0.0000e+00 |
| trans1x | 0.2162 | 0.0000e+00 |
| rgd | 0.2158 | 0.0000e+00 |
| orbnet_denali | 0.1711 | 1.2694e-243 |
| elytes | 0.0641 | 2.1944e-34 |
| metal_complexes | 0.0406 | 5.1420e-14 |
| biomolecules | 0.0183 | 3.4879e-03 |
| spice | 0.0078 | 6.0437e-01 |

## C.3    ROBUSTNESS OF $SO(3)$-UNIFORMITY TESTS

In App. C.2 we describe a Kolmogorov–Smirnov (KS) test to quantify the (non-)uniformity of molecular orientations. The KS test provides an alternative to the KL divergence presented as (non-)uniformity measure in App. C.1. To empirically study the robustness of both methods, we have subdivided $\sim 130,000$ samples from each dataset (QM9, QMugs, and OMol25) as well as 130,000 samples from a reference set of uniformly sampled rotations into $k$ (between 4 and 50) disjoint sets for various sizes $n$ and computed the KL divergences and KS test statistics on each subset. Box plots that illustrate the spread and biases for the different dataset sizes are shown in Fig. 9.

The KS test statistic seems to be a bit more robust, i.e. converges faster to a stable value with growing subset size. The reason that the QM9 dataset shows the strongest variation of the datasets is most likely due to the fact that the orientation distribution has substructure (non-uniformity) at a finer resolution which can only be resolved with a sufficiently large sample size.

Table 3: **Hyperparameters for training our simple message passing network on default orientation detection.**

| | Architecture hyperparameter |
|---|---|
| Bessel frequencies (radial) | 32 |
| Bessel frequencies (angular) | 20 |
| Num. message passing layers (Eq. (31)) | 3 |
| Aggregation operation in message passing | max |
| Node feature dimension | 512 |
| Hidden layers in message MLP | [128] |
| Activation function | SiLU |
| Radial cutoff for message passing | 10 Å |
| Hidden layers in readout MLP | [512, 128, 32] |

| | Training hyperparameter |
|---|---|
| Optimizer | AdamW |
| Weight decay | 5e-3 |
| Learning rate | 5e-4 |
| Scheduler | Cosine-LR |
| Epochs | 200 for OMol25, 100 otherwise |
| Warm-up epochs | 5 |
| Gradient clip | 0.5 |
| Loss function | BCE-loss |

## D  DETAILS REGARDING MODEL TRAINING AND ARCHITECTURES

**Details on the message passing architecture used for the detection of default orientations.**  In Sec. 3.1, we demonstrate that a simple geometric message passing network accurately discerns canonical samples from randomly rotated ones. For the model, we employ a straightforward point cloud architecture, consisting of three layers of message passing:

$$f_i^{(k+1)} = \bigoplus_{j \in \mathcal{N}(i)} \mathrm{MLP}(f_j^{(k)}, \mathrm{emb}(x_i - x_j)), \tag{31}$$

where $f_i^{(k)}$ denotes the feature vector of atom $i$ in layer $k$, $x_i$ its position and $\mathcal{N}(i)$ its neighborhood (defined by a radial cutoff of 10 Å). As aggregation function $\bigoplus_{j \in \mathcal{N}(i)}$ we use the component-wise max operation. The input to these message passing layers consists of a learned embedding of the neighbor geometry combined with an embedding of the atom type. For the initial node embedding, we use Bessel functions of the first kind with 32 learnable frequencies to embed the relative distance $r_{ij} = \|x_i - x_j\|_2$. Similarly, we use Bessel functions with 20 learnable frequencies to separately embed each component of the normalized relative distance vector as angular embedding. The aggregated (summed) angular and radial embeddings from the local neighborhood are combined with a one-hot embedding of the atom type to form the input node features $f_i^{(0)}$. During message passing, the angular and radial part of the relative distance vectors $x_i - x_j$ are embedded using 64 Gaussian radial basis functions (spaced equidistantly between 0 and 10 Å). The network is then trained to minimize a binary cross-entropy loss, predicting whether the input molecule has been randomly rotated or is in its (perturbed) default pose. All hyperparameters are summarized in Tab. 3.

**MLP used for the molecular property prediction from molecular orientations alone.**  In Sec. 3.3, we demonstrate that a simple MLP receiving as input only normalized principal components (PCs) of atom positions can successfully regress molecular properties. The MLP receives the first two normalized PCs as input, uses SiLU activations and four hidden channels with 256 features each. It is trained with MSE loss without weight decay. We have trained one separate model of the same architecture for each property from the different datasets reported in Tab. 1.

**Dealing with the dataset size.**  For QM9 we train all models for 100 epochs on the full dataset using a train-val-test split of (110,000, 10,000, ∼20,000). For the larger QMugs dataset we use a

train-val-test split of (80%, 10%, 10%). However, in order to keep the training time and learning rate scheduling comparable to the one in QM9, we train all QMugs models for 100 epochs with a different random training subsets of size 110000 for each epoch. For the massive OMol25 dataset we use a train-val-test split of (99.8%, 0.1%, 0.1%) and train for a total of 200 epochs again on different random subsets of the training set of size 110000.

# E ADDITIONAL EXPERIMENTS AND ABLATION STUDIES

## E.1 INVESTIGATING ORIENTATION BIAS IN ADDITIONAL DATASETS

As mentioned in Sec. 2.1, one subtle example where an orientation bias may easily be overlooked is the field of charge density prediction. In the field, several standard benchmarks exist that store DFT ground state electron densities on regular Cartesian (axis-aligned) grids as training targets (Brockherde et al., 2017; Bogojeski et al., 2020; Jørgensen & Bhowmik, 2022; Li et al., 2025). For the combined MD dataset Brockherde et al. (2017); Bogojeski et al. (2020), we visualize the distribution of orientations (as given by the principal components, cf. Sec. 3.2) in a Mollweide projection (Sec. 3.4) in Fig. 10. Additionally, we show the empirical distribution of relative angles between all orientations in the respective dataset and the most common orientation (described in Sec. 3.2).

The two other datasets also contain electron densities stored on regular Cartesian (axis-aligned) grids. They are both based on the standard QM9 dataset, which we have shown to exhibit strong orientation bias. Indeed, we could confirm that the molecular orientations are exactly the same as for the QM9 dataset presented in the main text.

Since several of the recently proposed approaches for relaxed equivariance train models on the molecular dynamics dataset rMD17 (Christensen & Von Lilienfeld, 2020), we have decided to incorporate it into our analysis (complementing the prior work by Lawrence et al. (2025a)). Figure 11 shows the plots equivalent to those described above for rMD17. We have used the rMD17 dataset available via `https://figshare.com/articles/dataset/Revised_MD17_dataset_rMD17_/12672038`. In particular, for the subsets of the rMD17 molecular dynamics dataset, it is surprising that the orientation distributions are qualitatively so different, given that all subsets should be generated in a comparable way.

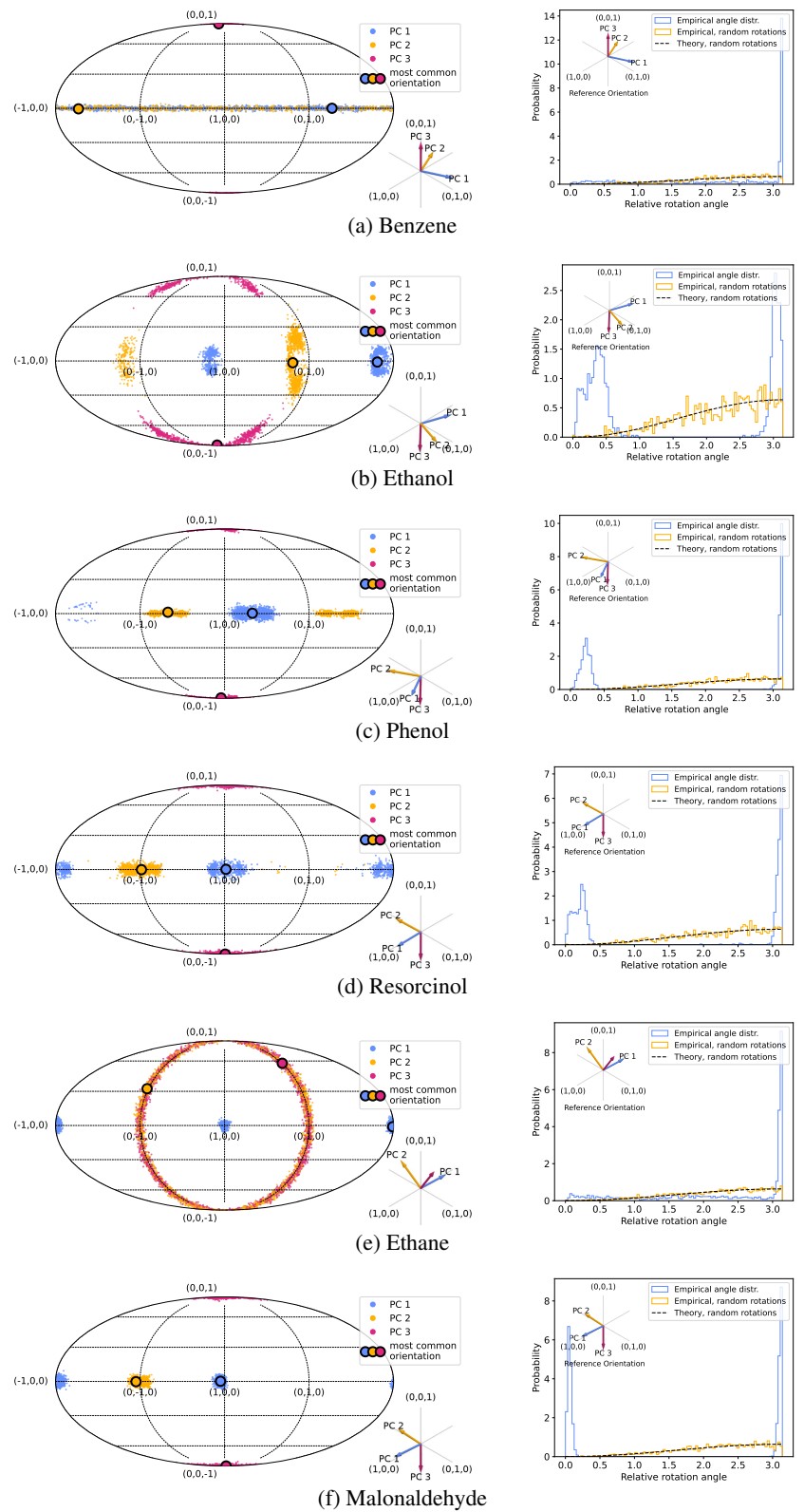

Figure 10: Mollweide projections of all molecular orientations (given by principal components) and distribution of relative rotation angles (relative to the most common orientation) for the MD electron density datasets of (Brockherde et al., 2017; Bogojeski et al., 2020), containing between 1000 and 2000 conformations for each molecule.

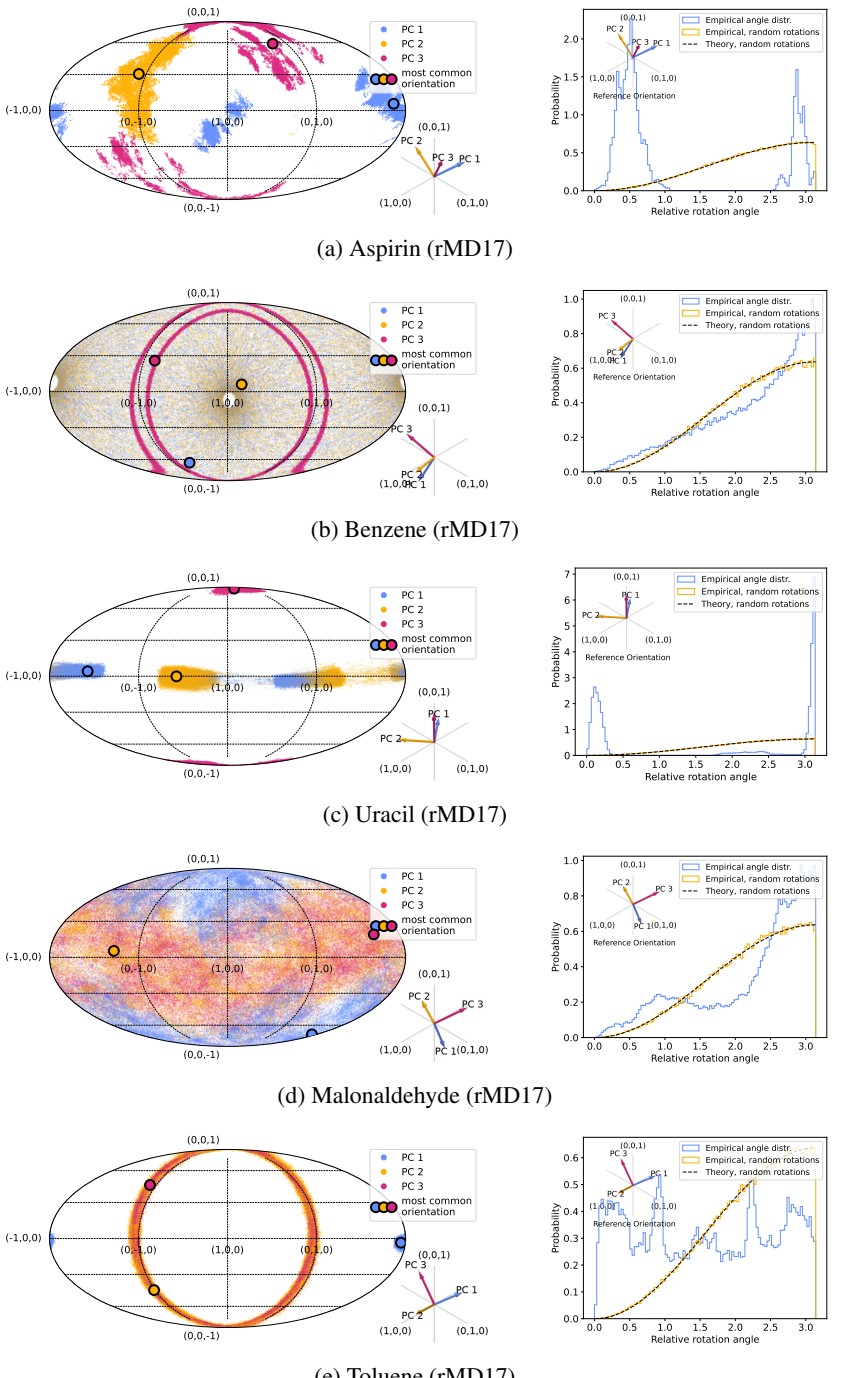

(a) Aspirin (rMD17)

(b) Benzene (rMD17)

(c) Uracil (rMD17)

(d) Malonaldehyde (rMD17)

(e) Toluene (rMD17)

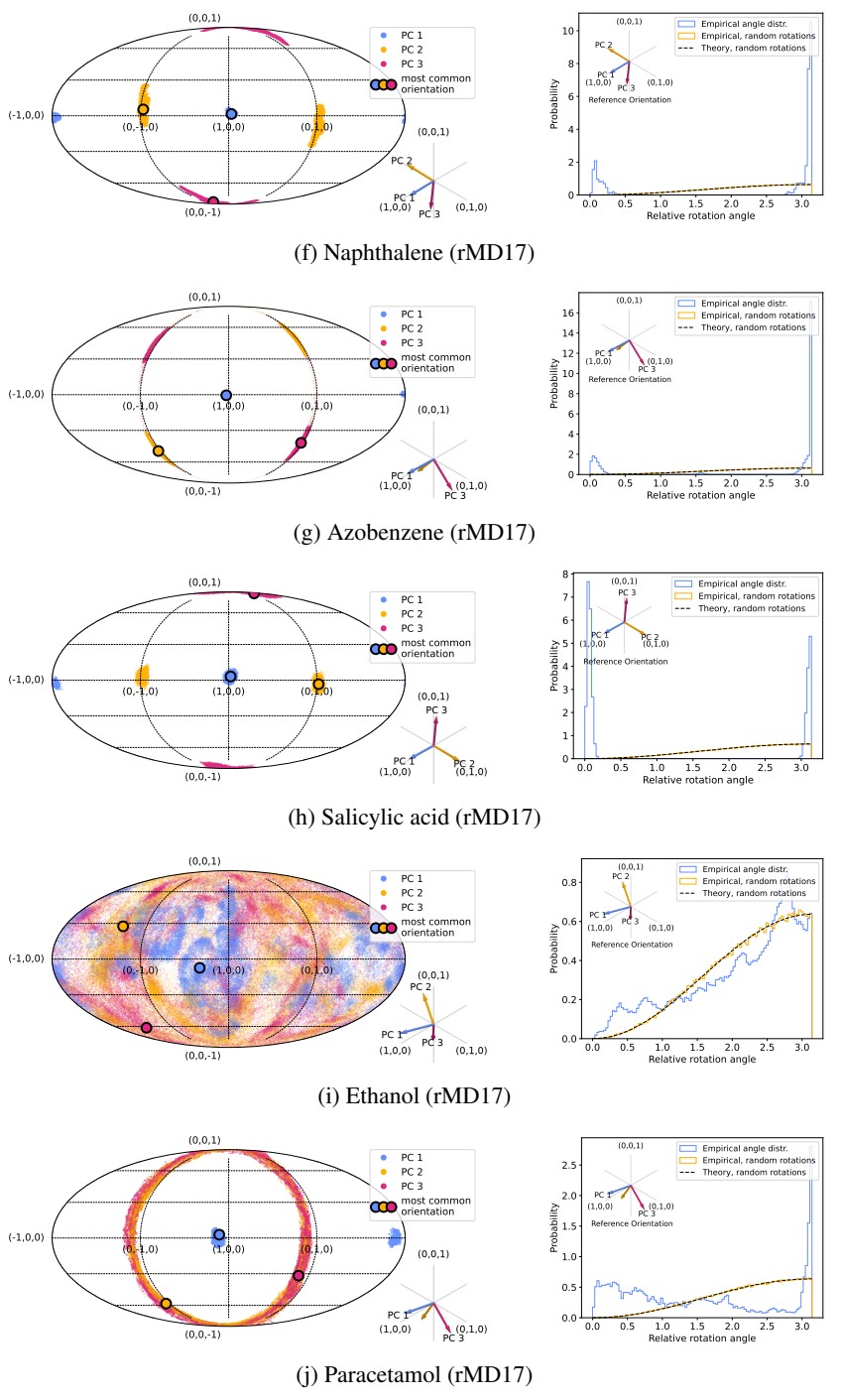

(f) Naphthalene (rMD17)

(g) Azobenzene (rMD17)

(h) Salicylic acid (rMD17)

(i) Ethanol (rMD17)

(j) Paracetamol (rMD17)

Figure 11: Mollweide projections of all molecular orientations (given by principal components) and distribution of relative rotation angles (relative to the most common orientation) for the rMD17 dataset (Christensen & Von Lilienfeld, 2020), containing 100,000 conformations for each molecule.

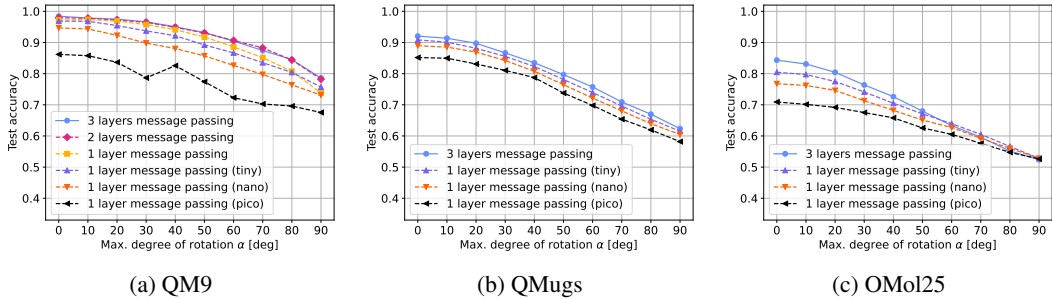

(a) QM9        (b) QMugs        (c) OMol25

Figure 12: Much smaller versions of the simple geometric message passing network can still accurately discern canonical samples from randomly rotated ones, even when pre-rotated by up to $\alpha$ before applying the random rotation to be detected. For each angle $\alpha$ we train a separate model. The 1-layer "pico" models only have 238 trainable parameters.

### E.2 DETECTING ORIENTATION BIAS WITH EVER SMALLER MESSAGE PASSING MODELS

The results presented in Fig. 3 show that a simple geometric message passing network can accurately discern canonical samples from randomly rotated ones. It is an interesting and relevant question to study the effect that shrinking the network size has on the classifier's accuracy. Reducing the number of trainable parameters lowers the training times and hardware requirements for probing a dataset for orientation bias. To that end, we have conducted an ablation study on QM9 by training a 2-layer and 1-layer message passing model. Both models still accurately discern the canonical dataset from a randomly rotated one (see Fig. 12). Further, we have drastically reduced the number of learnable parameters on the 1-layer network from 642,000 to a "tiny" version with 25,000, to a "nano" version with 2350, and to a "pico" version with only 238 trainable parameters. Very surprisingly, even the smallest versions only sacrifice a bit of accuracy but still adequately detect the orientation biases in all three datasets (see Fig. 12). This result seems to illustrate the remarkable inductive bias in the geometric message passing network and suggests that in practice the classifier test in Sec. 3.1 may be performed with a much smaller network.

### E.3 ADDITIONAL PROPERTY PREDICTION FROM A MOLECULE'S ORIENTATION ALONE

In Tab. 1 we have demonstrated, for a selection of molecular properties, that a simple MLP that is trained on property prediction on the canonical datasets and ingests *only* normalized principal components can significantly outperform the test performance achievable with uninformative input features. We have now additionally trained one separate MLP model to regress each (scalar) molecular property in the QM9 and QMugs datasets based solely on the principal components of the molecular geometries. In Tab. 4 and 5 we report the mean squared error (MSE) that the MLP achieves when trained on the standard datasets and on randomly rotated versions of them. We compare the MSE against the optimal test performance achievable for uninformative input features (as in Tab. 1). For (almost) all properties we find that the MLPs trained on the standard datasets achieve a significantly lower MSE than the mean of the test set. For several properties the relative improvements are comparable to those of the selected properties shown in Tab. 1. The improvement relative to the MSE of the mean of the test set is shown in percent and computed as $(MSE(\text{model}) - MSE(\text{test mean}))/MSE(\text{test mean})$. Since the standard deviations in Tab. 1 are comparatively small, we have conducted only one training for each property in this case.

### E.4 TRAINING AN MLP CLASSIFIER TO DISCERN ORIENTATIONS DIRECTLY

The results presented in Fig. 3 and 12 show that a simple geometric message passing network can accurately discern canonical samples from randomly rotated ones. The trained message passing networks receive as input the molecular geometry comprised of the atomic coordinates and the atom types. Since the distribution of orientations (given by the principal components) e.g. in Fig. 1 is far from uniform and contain non-trivial patterns, it is an interesting question how well a classifier can detect orientation bias from the principal eigenvectors directly. For that, we have trained vanilla MLPs ingesting only the molecule's orientation (in the form of the principal components) to predict

Table 4: **Molecular property prediction on QM9 from a molecule's orientation alone.** A simple MLP trained on the canonical datasets to regress molecular properties given *only* normalized principal components of atom positions as input outperforms the test performance achievable with uninformative input features on all properties. Shown in percent is the improvement of the MLP relative to the test mean.

| Property | MSE of MLP (w/ rot.) | MSE of mean (test) | MSE of MLP (w/o rot.) |
|---|---|---|---|
| $\mu$ [Debye] | 2.35 (0.11%) | 2.35 | 2.24 (-4.84%) |
| $\alpha$ [Bohr$^3$] | 66.40 (0.02%) | 66.38 | 63.54 (-4.29%) |
| $\epsilon_{HOMO}$ [eV] | 0.37 (0.02%) | 0.36 | 0.34 (-8.20%) |
| $\epsilon_{LUMO}$ [eV] | 1.64 (0.08%) | 1.64 | 1.42 (-12.88%) |
| $E_{\mathrm{gap}}$ [eV] | 1.66 (0.10%) | 1.66 | 1.40 (-15.72%) |
| $\langle R^2 \rangle$ [Bohr$^2$] | 75589.49 (0.02%) | 75573.68 | 67985.68 (-10.04%) |
| $E_{\mathrm{ZPVE}}$ [eV] | 0.81 (0.10%) | 0.81 | 0.62 (-23.09%) |
| $U_0$ [eV] | 1234359.12 (-0.01%) | 1234518.12 | 1209677.25 (-2.01%) |
| $U$ [eV] | 1234698.00 (0.02%) | 1234504.12 | 1209264.88 (-2.04%) |
| $H$ [eV] | 1234502.12 (-0.00%) | 1234504.38 | 1209401.00 (-2.03%) |
| $G$ [eV] | 1234789.88 (0.02%) | 1234547.75 | 1210481.62 (-1.95%) |
| $C_v$ [cal mol$^{-1}$ K$^{-1}$] | 16.17 (0.02%) | 16.17 | 13.79 (-14.71%) |
| $U_0^{\mathrm{atom}}$ [eV] | 106.79 (0.05%) | 106.73 | 87.31 (-18.20%) |
| $U^{\mathrm{atom}}$ [eV] | 108.68 (0.04%) | 108.64 | 88.33 (-18.69%) |
| $H^{\mathrm{atom}}$ [eV] | 110.23 (0.04%) | 110.18 | 89.26 (-18.99%) |
| $G^{\mathrm{atom}}$ [eV] | 90.41 (0.03%) | 90.39 | 73.87 (-18.28%) |
| $A$ [GHz] | 10.18 (0.02%) | 10.18 | 9.81 (-3.58%) |
| $B$ [GHz] | 8.20 (0.01%) | 8.20 | 8.16 (-0.43%) |
| $C$ [GHz] | 3.56 (0.01%) | 3.56 | 3.54 (-0.46%) |

Table 5: **Molecular property prediction on QMugs from a molecule's orientation alone.** A simple MLP trained on the canonical datasets to regress molecular properties given *only* normalized principal components of atom positions as input outperforms the test performance achievable with uninformative input features on most properties. Shown in percent is the improvement of the MLP relative to the test mean.

| Property | MSE of MLP (w/ rot.) | MSE of mean (test) | MSE of MLP (w/o rot.) |
|---|---|---|---|
| GFN2:TOTAL_ENERGY | 890.55 (0.01%) | 890.48 | 843.62 (-5.26%) |
| GFN2:ATOMIC_ENERGY | 653.21 (0.01%) | 653.17 | 618.43 (-5.32%) |
| GFN2:FORMATION_ENERGY | 19.69 (0.01%) | 19.69 | 18.76 (-4.72%) |
| GFN2:TOTAL_ENTHALPY | 880.71 (0.01%) | 880.66 | 834.10 (-5.29%) |
| GFN2:TOTAL_FREE_ENERGY | 882.03 (0.01%) | 881.91 | 835.28 (-5.29%) |
| GFN2:HOMO_ENERGY | 0.0003285 (0.0031%) | 0.0003285 | 0.0003285 (-0.0013%) |
| GFN2:LUMO_ENERGY | 0.0007593 (0.0068%) | 0.0007593 | 0.0007592 (-0.0034%) |
| GFN2:HOMO_LUMO_GAP | 0.0007008 (0.0038%) | 0.0007007 | 0.0007006 (-0.0178%) |
| GFN2:FERMI_LEVEL | 0.0003687 (0.0099%) | 0.0003687 | 0.0003687 (0.0004%) |
| GFN2:DISPERSION_COEFFICIENT_MOLECULAR | 1035579456 (0.02%) | 1035421632 | 997401536 (-3.67%) |
| GFN2:POLARIZABILITY_MOLECULAR | 8677.71 (0.01%) | 8677.08 | 8224.04 (-5.22%) |
| DFT:TOTAL_ENERGY | 618583 (0.01%) | 618508 | 603139 (-2.48%) |
| DFT:ATOMIC_ENERGY | 616063 (0.00%) | 616037 | 600839 (-2.47%) |
| DFT:FORMATION_ENERGY | 11.37 (0.01%) | 11.37 | 10.85 (-4.59%) |
| DFT:XC_ENERGY | 2287.17 (0.01%) | 2287.04 | 2169.66 (-5.13%) |
| DFT:NUCLEAR_REPULSION_ENERGY | 3938205 (0.01%) | 3937772 | 3695442 (-6.15%) |
| DFT:ONE_ELECTRON_ENERGY | 22159716 (0.01%) | 22157832 | 20835828 (-5.97%) |
| DFT:TWO_ELECTRON_ENERGY | 4894430 (0.01%) | 4894038 | 4595915 (-6.09%) |
| DFT:HOMO_ENERGY | 0.0003445 (0.0074%) | 0.0003445 | 0.0003438 (-0.2061%) |
| DFT:LUMO_ENERGY | 0.0007075 (0.0054%) | 0.0007075 | 0.0007056 (-0.2714%) |
| DFT:HOMO_LUMO_GAP | 0.0009014 (0.0055%) | 0.0009013 | 0.0008966 (-0.5185%) |

whether the PCs come from a previously rotated molecular geometry or not. Conceptually, this test is similar to our non-parametric uniformity test (discussed in Sec. 3.2 and App. C.1). Indeed, the MLP classifier can still distinguish well between samples from the two distributions (as shown in Fig. 13). However, since the principal components arguably contain fewer cues about a molecule's orientation than the full geometry, it seems plausible that the classifier is overall less accurate than the message passing network. For instance, in some cases the canonical orientation may also be detectable from the orientation of local substructure in the molecule – information which is not contained in the PCs.

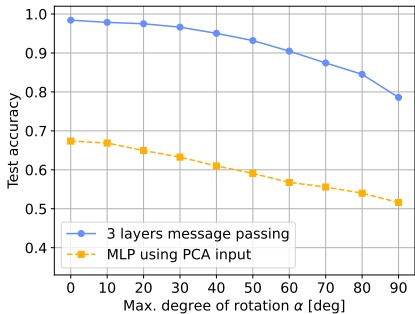

Figure 13: A vanilla MLP ingesting only the molecule's orientation (in the form of the principal components) can still predict whether the PCs come from a previously rotated molecular geometry or not (dataset: QM9). Similar to the experiments presented in Fig. 3 and 12 the canonical orientations were previously randomly rotated by an angle up to $\alpha$ to examine the robustness of the detection. Overall, the MLP performs worse than a geometric message passing network which receives the molecular geometry as input. The reason for that may simply be that the molecular geometry contains more cues about the exact orientation of a molecule.

### E.5 STUDYING THE EMPIRICAL EFFECT OF ORIENTATION BIAS ON NNS

Beyond the experiments reported in Tab. 1, 4 and 5, it is an interesting but involved research question to study which type of architectures and on which tasks (e.g. prediction of scalar quantities vs. tensorial quantities) are how strongly affected by orientation bias. As a first step, we have implemented an adaptation of the Graphormer architecture (Ying et al., 2021) to study the effect of orientation bias on a typical transformer model. Indeed, despite its simplicity, the Graphormer is still actively used in molecular ML Zhang et al. (2024); Remme et al. (2025). Our adaptation uses a geometric embedding similar to the one used for the message passing network (cf. App. D), followed by 4 Graphormer layers (32 heads, 512 channels) in which a radial (and optionally an angular) embedding are used as offsets on the attention weights. By default, the Graphormer is rotationally-invariant and only uses radial embeddings of relative positions in the attention mechanism. However, for comparison, we optionally modify the architecture to further ingest a simple angular embedding of the normalized relative distance vectors $\frac{x_i - x_j}{\|x_i - x_j\|}$. Importantly, this naive angular embedding breaks the exact invariance of the architecture. We have trained the invariant and non-invariant version of this model on molecular property prediction using a) the canonical QM9 and QMugs dataset and b) dataset versions in which orientations were previously randomized (see Tab. 6). All models were trained with L1-loss for 100 epochs (learning rate: $5 \times 10^{-4}$ and weight decay $5 \times 10^{-3}$). As molecular properties we have chosen the ZPVE for QM9 and the DFT:NUCLEAR_REPULSION_ENERGY for QMugs, since these showed the strongest relative improvements in Tab. 4 and 5 respectively.

As expected, in all cases, we find that the non-equivariant models trained on the canonical data strongly rely on the alignment of orientations. When evaluating these models on the datasets with randomized orientations the error metrics (mean absolute error and mean squared error) increase significantly. For the QMugs property, it seems that the orientation-randomized and thus more varied version of the data help the model to learn more informative patterns (comparable to the effect of data augmentation). Here, the model trained on randomized data achieves a better performance than the model trained on canonicalized data, even when evaluating on canonicalized data. However, highlighted in bold (QM9, MSE) we see a scenario in which the opposite is the case and the model trained on canonicalized data also outperforms the model trained on randomized orientations. This

is the critical case, in which a non-equivariant model leverages the orientation bias in the dataset to artificially inflate its performance. Further, we find that for both properties (except in the MAE metric for the QM9 property) the fully-invariant model performs best. A toy example in which the canonicalization should show a strong effect is the regression of the first principal component, given the molecular geometry as input (see Tab. 6c). Indeed, in this scenario, the non-equivariant Graphormer trained on the canonical dataset leverages the orientation bias in the canonical dataset to outperform the same model trained on randomized orientations. This experiment is intended to serve as a proxy for the prediction of vectorial quantities (such as the dipole moment) which could be highly correlated with the orientation of a molecule.

All in all, it is very reasonable to assume that the effect of the orientation bias on model performance strongly varies depending on the architecture, the dataset size and the nature of the molecular target.

Table 6: Invariant and non-invariant Graphormer adaptations trained and evaluated on molecular property prediction with and without canonicalized molecular orientations.

(a) Regression of ZPVE [eV] on QM9

| train \ test | MAE canonical | MAE randomized | MSE canonical | MSE randomized |
|---|---|---|---|---|
| canonical | 0.0049 | 0.0051 | **0.000045** | 0.000048 |
| randomized | 0.0031 | 0.0031 | 0.000062 | 0.000062 |
| invariant model | 0.0036 | 0.0036 | 0.000027 | 0.000027 |

(b) Regression of DFT:NUCLEAR_REPULSION_ENERGY [$E_h$] on QMugs

| train \ test | MAE canonical | MAE randomized | MSE canonical | MSE randomized |
|---|---|---|---|---|
| canonical | 107.2 | 150.8 | 77127 | 111421.054688 |
| randomized | 85.7 | 85.8 | 66544 | 66800 |
| invariant model | 63.6 | 63.6 | 51237 | 51237 |

(c) Regression of PC1 on QMugs

| train \ test | MAE canonical | MAE randomized | MSE canonical | MSE randomized |
|---|---|---|---|---|
| canonical | **0.3556** | 0.6296 | **0.3442** | 0.6190 |
| randomized | 0.3569 | 0.3565 | 0.3604 | 0.3597 |

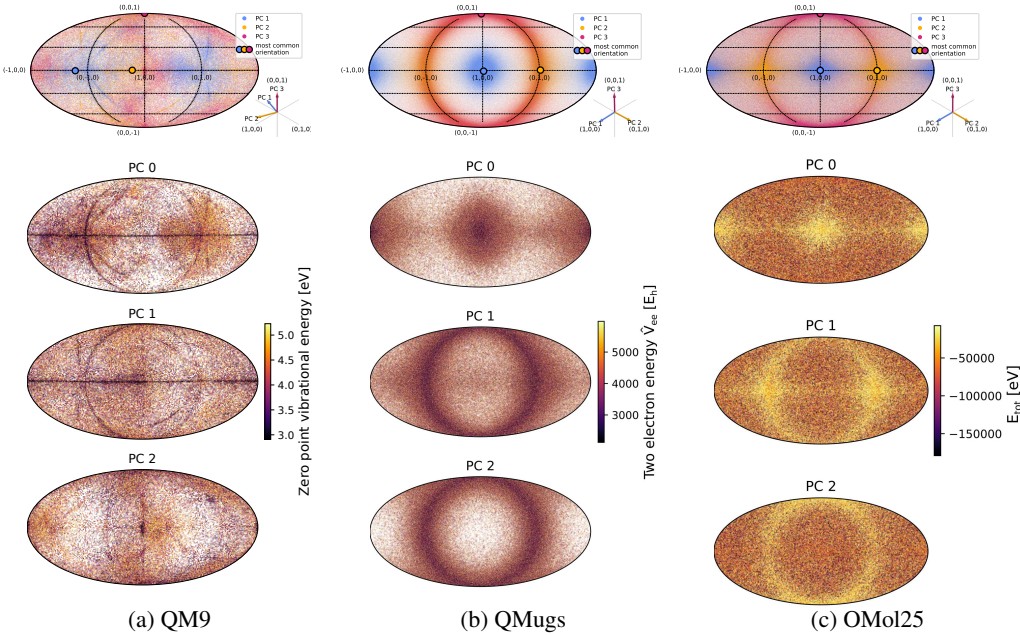

(a) QM9        (b) QMugs        (c) OMol25

Figure 14: Combined Mollweide projections of all molecular orientations (given by principal components) and molecular properties. Each orientation consists of a triplet of points in the 2D projection. In the top row, the triplets are plotted jointly (blue $\sim$ PC1, yellow $\sim$ PC2, magenta $\sim$ PC3). The plots below show the distribution of each PC separately colored by chemical properties. The substructure reveals the correlation between canonical orientations and chemical properties.

## E.6 Combined visualization of molecular orientations and molecular properties

For completeness, we present a combined visualization of all molecular orientations and the corresponding property-colored visualizations to reveal correlations between orientations and molecular properties (Fig. 14).

**LLM usage.** Large Language Models (LLMs) were used in the preparation of this submission to polish the writing regarding formulations and wording. In addition, we have used LLM-based auto-completion in the development of our research code.

