# OpenReview forum: "Take Note: Your Molecular Dataset Is Probably Aligned"
_ICLR.cc/2026/Conference — ICLR 2026 Poster_

### Official Review · Reviewer_F9xk · 2025-10-17

**Soundness:** 2
**Presentation:** 2
**Contribution:** 2
**Rating:** 4
**Confidence:** 3

**Summary:**

The paper describes the inherent bias present in the open-source molecular datasets (QM9, QMugs, and OMol25) in terms of molecular poses when applied to the task of predicting properties of these molecules.

**Strengths:**

- The issue of dataset-induced bias in molecular ML is novel, especially as orientation-invariant architectures (e.g., SE(3)/E(3)-equivariant networks) are increasingly used.
- The Mollweide projections and density analyses effectively highlight the non-uniformity of molecular orientations.

**Weaknesses:**

- I find the paper’s argument about “pose bias” somewhat unclear. The 3D molecular geometries in datasets like QM9, QMugs, and OMol25 are generated using physics-based computational chemistry software, and therefore correspond to energetically stable conformers of each molecule. These conformations are naturally determined by the molecule’s 2D topology, atom types, and underlying energy landscape. From a physical standpoint, certain configurations or poses are expected to be more stable and thus more frequently observed — similar to how structures in the Protein Data Bank (PDB) reflect biologically preferred conformations. Given this, it seems there may be a misinterpretation between “pose bias” (a coordinate or orientation artifact) and physical realism (the fact that molecules adopt specific stable conformations). Could the authors clarify whether the observed “pose bias” truly represents an artifact of how datasets are stored and oriented, or whether it simply reflects physically meaningful conformational preferences inherent to molecular systems?

- I understand that, when developing methods for downstream applications on these molecular datasets, it is important to account for how global molecular poses relate to their representation in Cartesian coordinates. However, most state-of-the-art molecular machine learning architectures are E(3)-equivariant or invariant, meaning they operate on geometric quantities such as interatomic distances, angles, and relative orientations, rather than on absolute Cartesian coordinates. Consequently, these models are inherently insensitive to global rotations and translations of the molecule. This raises the question of how practically relevant the reported orientation bias is, given that such equivariant models should, by design, be unaffected by the dataset’s global coordinate frame.

- I think it would be more beneficial to check if this so-called "pose bias" is demonstrated in the models that have been trained on these datasets such as EGNNs, etc, to see if they are also susceptible to the variations wrt random rotations.

**Questions:**

- Can the authors describe how the random rotation operation is being executed in Sec 3.3, as it's not fully clear how it is applied to the canonical molecular pose? Is it a global or local transformation applied to the canonical position?

---

> ### Author Response · Authors · 2025-11-22
> **Answer to Reviewer F9xk**
>
> Thank you for your detailed and constructive review. We are happy to hear that you find our investigations novel and our visualizations and density analyses convincing. Thanks as well for the questions and criticism, which we address in the following.
>
> ### 1 Orientation bias vs. physical geometry optimization
>
> > Could the authors clarify whether the observed "pose bias" truly represents an artifact of how datasets are stored and oriented, or whether it simply reflects physically meaningful conformational preferences inherent to molecular systems?
>
> Throughout our paper, when talking about the orientation of a molecule, we refer to its global orientation. We characterize the orientation of each molecule by a single rotation matrix as described in the beginning of Sec. 3.2. Without an external potential, the global orientation of a molecule is arbitrary, i.e. can be chosen freely while respecting all physically meaningful bond lengths and angles. Therefore, in principle, every orientation is equally valid. Still, cheminformatics tools, which generate 3D molecular geometries (e.g. from SMILES strings), explicitly choose a global orientation when optimizing relative atom positions to relax the geometry. In our work, we demonstrate that these global orientations are (often) chosen in a biased way and highlight this as an artifact of the data generating process. While we believe that cheminformaticians are well aware of this fact, it can still be a real pitfall for machine learning practitioners. In our text we use the term "pose" as a synonym to (global) orientation.
>
> ### 2 Why detecting orientation bias is relevant
>
> > This raises the question of how practically relevant the reported orientation bias is, given that such equivariant models should, by design, be unaffected by the dataset's global coordinate frame.
>
> We fully agree with the reviewer that rotationally-equivariant models are agnostic to the global orientation of molecular geometries by design, as stated in l. 036f.
> However, due to the limitations that equivariance imposes on the architectural design and due to the challenges related to the optimization of networks with built-in equivariance, more and more molecular ML approaches are based on relaxed or (learned) approximate equivariance. These networks are susceptible to orientation bias (see answer 6 to Reviewer YdZo for details).
>
> Furthermore, as stated in l.086f, when molecular properties such as the ground state electron density are evaluated on regular Cartesian grids that are not spherically symmetric (Brockherde et al., 2017; Bogojeski et al., 2020; Jorgensen & Bhowmik, 2022; Li et al., 2025), the canonical orientations may introduce a systematic bias even when equivariant architectures are used later in the prediction (see answer 6 to Reviewer pzKZ for details).
>
> In these more subtle cases, the orientation bias may easily be overlooked and can lead to undesired effects.
>
> ### 3 Testing orientation bias on trained models
>
> > I think it would be more beneficial to check if this so-called "pose bias" is demonstrated in the models that have been trained on these datasets such as EGNNs, etc, to see if they are also susceptible to the variations wrt random rotations.
>
> As stated above, fully equivariant models (such as EGNNs) are not affected from orientational bias in geometric datasets. For non-equivariant models, we agree with the reviewer that it is an interesting but involved research question to study how much the performance of published architectures is affected by orientation bias in molecular datasets. As a first step in this direction, we have trained a Graphormer model in the different settings (see answer 7 to Reviewer YdZo for more information).
>
> ### 4 Random rotations in Sec. 3.3
>
> > Can the authors describe how the random rotation operation is being executed in Sec 3.3, as it's not fully clear how it is applied to the canonical molecular pose? Is it a global or local transformation applied to the canonical position?
>
> For the experiments described in Sec. 3.3 we apply one global transformation on each molecule that jointly rotates all atom positions, equivalent to the rotation described in Eq. (2). The rotations differ from molecule to molecule and are sampled uniformly from the set of all rotations $\mathrm{SO}(3)$. Additionally, we use deterministic rotations conditioned on the molecule's index so that the same sample will be transformed with the same rotation when revisited during training. This is equivalent to using a version of the dataset in which each molecule has been rotated once prior to training.
>
> We hope we were able to address your questions and look forward to discussing further.

---

### Official Review · Reviewer_pzKZ · 2025-10-30

**Soundness:** 4
**Presentation:** 3
**Contribution:** 4
**Rating:** 10
**Confidence:** 5

**Summary:**

This paper provides clear evidence of subtle leakage in major molecular conformation datasets used for public comparisons of machine learning applications to drug discovery. Notably, the leakage arises because molecules with similar properties have similar geometric orientations in these datasets. The authors demonstrate this defect with increasing levels of rigor: from simple visualization, to discriminating rotated vs unrotated molecules with a simple classifier (even in the presence of substantial noise), to comparing the densities of orientational distributions.  Importantly, the paper shows that simply reading a molecule's orientation enables property prediction with a substantial reduction in variance compared to the test set mean. Finally, the paper includes equal-area Mollweide projections of the overall orientational distributions in small molecule datasets that show the relative degree of the lack of true randomness across datasets.

**Strengths:**

This paper is original and significant because it analyzes a subtle effect that might prematurely and incorrectly influence the selection of models based on their observed performance. It is particularly timely, as limitations imposed from forced SE3 equivariance in existing architectures could be replaced by learned equivariance with faster/stronger architectures and larger datasets. Unfortunately, careful preprocessing of the standardized datasets is often avoided, but the magnitude of the effects shown in this paper clarify that such practice is unacceptable going forward. The quality of the work is high; as a subtle example: the authors apply single fixed rotations to un-randomized molecules when demonstrating their orientation-based property predictors to minimize any accidental deviation in their data pipelines. The flow of the presentation is clear and the message is strong. Duly Noted!

**Weaknesses:**

The main weakness is the lack of accompanying code for reproducibility, which should be resolved prior to publication. The lack of code would unfairly reduce the applicability of this work in future dataset preparations and benchmarks (although the implementations appear simple, rotations warrant care.)  A minor weakness is the hard-to-discern densities in some combined Mollweide plots.  While the lack of uniformity across datasets makes them memorable, comparing Fig 5 to 6 I wonder if the order of plotting the dots in Fig 6 is deterministic, rather than random, and hides information. A suggestion: perhaps also try a more distinct/color-blind friendly color palette (#0072B2 #E69F00 #CC79A7) and add a combined QM9 plot in the supplement for completeness.

**Questions:**

Due to the possibility of reflection symmetries in small molecules (think small symmetric fragments), exact degeneracies can arise from their geometries. How do the authors handle degeneracies in the ranking of the principal components or the max function in Eq 3?

Fig 4 convincing and clear. I'd recommend that future datasets randomize orientations and include this analysis and the Mollweide equal-area plots.  Did the authors perhaps also try to use a simple summary metric of Fig 4, e.g. the distance between the actual vs intended CDE of the distributions?  Would the ranking of such a summary metric agree with the visual ranking in Fig 7?

Does this work imply issues in any specific prior publications that used symmetry-breaking, approximate-equivariant, or non-equivariant methods that relied on experimental evaluations? Highlighting any suspicious references (even in supplements or the openreview discussions) could benefit the community, though I understand that it's not the paper's main focus and nobody likes to be that guy.

---

> ### Author Response · Authors · 2025-11-22
> **Answer to Reviewer pzKZ**
>
> Thank you for your thorough and constructive review. We are very happy to hear that you find the quality of our work high (both in presentation and substance) and we much appreciate your endorsement to share our findings with the community. Thanks as well for the questions and criticism, which we address in the following.
>
> ### 1 Publication of code
>
> > The main weakness is the lack of accompanying code for reproducibility, which should be resolved prior to publication.
>
> We strongly agree with the reviewer that our code should serve as a valuable resource to the community and should be usable at a low entry barrier. Upon publication, we will make our code available to reproduce all presented experiments. We will make sure that our code is readily applicable for the preparation and curation of future datasets.
>
> ### 2 Randomized scattering in Mollweide plots
>
> >A minor weakness is the hard-to-discern densities in some combined Mollweide plots. While the lack of uniformity across datasets makes them memorable, comparing Fig 5 to 6 I wonder if the order of plotting the dots in Fig 6 is deterministic, rather than random, and hides information.
>
> We have double-checked that the plotting order of dots in all Mollweide scatter plots is indeed random, also across different PCs for the joint plots.
> In case the reviewer was wondering why in the joint plots (Fig. 5) the discrepancies between the distributions of PCs 2 and 3 are much clearer compared to Fig. 6, let us add the following explanation:
> In both plots, many points overlap in the very dense regions such that only a subset of points, the ones plotted late, is visible there. In Fig. 5, where PCs are plotted jointly, the fraction of e.g. yellow points in such dense regions equals (in expectation) the ratio of density of the distribution of PC1 relative to the density of the other two PCs.
> Hence, the joint plot emphasized differences in the distributions of the PCs more strongly. Furthermore, the granularity of the two plots is perceived as qualitatively different since we use higher marker size and opacity in Fig. 6 to highlight the dependence of properties on the orientation rather than the inhomogeneity of the distribution of poses.
>
> ### 3 Color-blind friendly color palette
>
> > A suggestion: perhaps also try a more distinct/color-blind friendly color palette (#0072B2 #E69F00 #CC79A7) and add a combined QM9 plot in the supplement for completeness.
>
> Throughout the paper we have used the IBM color-blind friendly color palette (https://lospec.com/palette-list/ibm-color-blind-safe), since we personally like its modern style and were hoping that the colors are well-distinguishable.
>
> Regarding the "combined QM9 plot", we are not exactly sure what the reviewer meant. Perhaps, the reviewer was looking for the Mollweide projection of all orientations in the QM9 dataset, which is shown in Fig. 1, equivalent to the visualizations for QMugs and OMol25 in Fig. 5. Please indicate if actually something else was meant.
>
> ### 4 Breaking ties in the computation of principal components
>
> > Due to the possibility of reflection symmetries in small molecules (think small symmetric fragments), exact degeneracies can arise from their geometries. How do the authors handle degeneracies in the ranking of the principal components or the max function in Eq 3?
>
> For molecules with exact symmetries (equally large PCA eigenvalues or reflection symmetry w.r.t. Eq. (3)) we break ties randomly. Since these are only a handful of molecules, we do not expect these cases to strongly effect the overall analysis.
>
> ### 5 One summary metric for non-uniformity
>
> > Did the authors perhaps also try to use a simple summary metric of Fig 4, e.g. the distance between the actual vs intended CDE of the distributions? Would the ranking of such a summary metric agree with the visual ranking in Fig 7?
>
> Thanks for the suggestion. During the rebuttal we have devised an additional non-parametric statistical test to measure $\mathrm{SO}(3)$-uniformity in a single number. Our measure is based on the Kullback–Leibler divergence between the empirical distribution of molecular orientations and the uniform distribution of rotations. Please take a look at answer 1 to Reviewer wjTX for more information.

---

> > ### Author Response · Authors · 2025-11-22
> > **Answer to Reviewer pzKZ (continued)**
> >
> > ### 6 Prior work that may be affected by orientation bias
> >
> > > Does this work imply issues in any specific prior publications that used symmetry-breaking, approximate-equivariant, or non-equivariant methods that relied on experimental evaluations?
> >
> > One subtle example where an orientation bias may easily be overlooked is the field of charge density prediction. In the field, several standard benchmarks exist that store DFT ground state electron densities on regular Cartesian (axis-aligned) grids as training targets [1-4].
> >
> > Our analysis demonstrates that indeed all four datasets are highly canonicalized. The combined datasets of [1,2] contain molecular dynamics trajectories for six different molecules (around 1000 data samples each). For each subset we visualize all molecular orientations in Fig. 9 of the revised manuscript. The datasets [3,4] both contain electron densities evaluated on regular Cartesian grids for all QM9 molecules. The latter two datasets differ in the DFT software used to calculate the reference electron density: VASP in Ref. [3] and PySCF in Ref. [4]. Both datasets [3,4] use the standard QM9 dataset with orientation bias. Since the volumetric grids are not spherically symmetric, the evaluation of the loss function is not rotationally-invariant which may introduce a systematic bias even when equivariant architectures are used later in the prediction. We can think of two scenarios in which such bias could be a problem:
> > 1) When practitioners deploy trained models, e.g. in MD simulations, they will (most likely) not rotate the simulated molecules in a preferred orientation, which in turn possibly decreases the performance of the model.
> > 2) When studying extrapolation capabilities of trained models, e.g. for larger molecules, the generalization to new datasets may be hindered by orientation biases in the datasets.
> >
> > Additionally, please take a look at answer 6 to Reviewer YdZo for examples of prior works that use relaxed or learned equivariance. Both aspects are now presented in an additional section 2.1 as part of the related work.
> >
> > [1] Felix Brockherde, Leslie Vogt, Li Li, et al. Bypassing the Kohn-Sham equations with machine learning. Nature Communications, 8(1):872, October 2017. https://www. nature.com/articles/s41467-017-00839-3.
> > [2] Mihail Bogojeski, Leslie Vogt-Maranto, Mark E. Tuckerman, et al. Quantum chemical accuracy from density functional approximations via machine learning. Nature Communications, 11(1):5223, October 2020. https://www.nature.com/articles/s41467-020-19093-1.
> > [3] Peter Bjørn Jørgensen and Arghya Bhowmik. Equivariant graph neural networks for fast electron density estimation of molecules, liquids, and solids. npj Computational Materials, 8(1):183, 2022.
> > [4] Chenghan Li, Or Sharir, Shunyue Yuan, and Garnet Kin-Lic Chan. Image super-resolution inspired electron density prediction. Nature Communications, 16(1):4811, 2025.
> >
> > We hope we were able to address your questions and look forward to discussing further.

---

> > ### Comment · Reviewer_pzKZ · 2025-11-22
> > **Thanks for the answers**
> >
> > I look forward to your release of the codes.
> >
> > Regarding the visualization of Fig 1 vs 5, from the current plot it was unclear if the scales/colors were the same, but I understand now that they are.  To ease comparison between figures 5 and 6, it may be useful to paste an addition row to Fig 6 with the 3 plots from Figs 1 and 5, and such a figure could be supplemental.
> >
> > Regarding the comparison of the distributions, although KL is a decent start and it seems to work fine on your case compared to the empirical ordering, I'd personally prefer a robust metric based on the cumulative distribution function estimates, something like a Wasserstein distance, or the KS.  It is not a major point, but estimates based on the PDF rather than the CDF, tend to lack robustness, which may become important if the datasets are small or have specific outliers.

---

> > > ### Author Response · Authors · 2025-11-25
> > > **Additional KS-test for SO(3)-uniformity**
> > >
> > > Many thanks for your reply. We appreciate your continued interest in the comparison of the distribution of orientations. Following your suggestion, we have devised a Kolmogorov–Smirnov test to compare the empirical distance distribution between orientations and the expected reference given by Eq. (5). We have studied the robustness of both (non-)uniformity tests and find that the KS test indeed gives a slightly more robust estimate. Please take a look at App. F for details. We have collected the material in an additional appendix to keep line numbers and figure references consistent with the already provided answers. We will properly include the material in the next revision.

---

### Official Review · Reviewer_YdZo · 2025-10-31

**Soundness:** 2
**Presentation:** 2
**Contribution:** 2
**Rating:** 2
**Confidence:** 4

**Summary:**

The paper studies the prevalence of canonicalized poses in commonly used molecular datasets (QM9, QMugs, OMol25). The authors use a simple classification test to distinguish between molecules in their original orientation in the dataset and ones that are randomly rotated. They also present visualization of principal components of molecular orientations for each dataset and the distributions of relative rotation angles between molecules. Finally, they demonstrate that one can use the orientation alone to regress on molecular properties and that this achieves better performance than using randomly rotated features.

**Strengths:**

The paper is well written, organized, and easy to follow.The experiment with property regression from molecular orientation is quite interesting and illuminates that one should be cautious when using non-equivariant models (the model could memorize spurious information contained in the molecular orientation). The visualizations of correlations of molecular orientation with chemical properties (Figure 6) are also quite interesting. In general, the paper makes a good point that one should be cautious about data pre-processing and pre-existing canonicalizations in the data.

**Weaknesses:**

A paper at the ICLR 2025 AI4Mat workshop (posted Mar 3, 2025) essentially presented the same idea [1] of a two-sample classifier test to detect whether a dataset is canonicalized or not, also applied to molecular datasets. [1] introduced the idea that these canonicalization biases may exist in molecular datasets. Furthermore, [1] showed that orientation of molecules in commonly used materials science benchmarking datasets is non-uniform (applied to QM9, MD17, OC20) and suggested that non-equivariant models may be strongly benefiting from canonicalization of the molecules’ orientations. This work seems to be quite similar, so I am not sure of the novelty ([1] also visualizes the distribution of principal moments of inertia for QM9 in the appendix). The framing and setup of [1] is very similar, but the authors of this work do not cite or mention this prior work. [1] also provided a more detailed treatment of the classifier test, showing how to compute p-values for a test of level $\alpha$ and validation of classifier performance, which this paper does not discuss. The authors should clarify the novelty of their work relative to [1].

Additionally, in relation to section 3.1, two-sample classifier tests have already been presented in the literature, so I believe [2] should be cited. There also exists literature on non-parametric hypothesis tests for distributional group symmetry and statistical tests for invariance [3,4], so these works should be cited or discussed.

While the visualization of molecular principal components are quite aesthetically pleasing, I am also not sure of their novelty. It is reasonable to expect that a machine learning practitioner would perform PCA on their data or visualize certain molecules prior to training a model, but much of the paper seems to rely on these visualizations. For example, take the observation that “chemically similar molecules tend to have similar canonical orientations.” For QM9, this most likely follows directly from the way conformers are generated (via CORINA from SMILES strings). Since CORINA uses deterministic geometry generation rules, this would be expected (the authors also state this). Therefore, I am not sure of the novelty of this finding/if it needs to be presented in the main part of the paper. For the other datasets (QMugs and OMol25) the authors should add further information as to why they think it has an orientational bias (L155).

From my understanding, equivariant models generally perform best for molecular property regression on many of these datasets (outperforming non-equivariant models e.g. https://benchmarks.rowansci.com/). If there is useful information contained in the molecular orientations, shouldn’t we expect the non-invariant models to do better? There are no experiments comparing equivariant, non-invariant, or approximately equivariant models to support the claims such as “architectures that introduce symmetry breaking or are only approximately equivariant…might artificially inflate their performance by exploiting the extrinsic canonicalization” (L082). Thus, it is hard to draw meaningful conclusions. To strengthen the paper, it would be good to have more experiments comparing models on molecular property prediction.

The test in Section 3.3 is interesting, but the claim it is “empirical proof that canonical orientation alone holds information about a molecule’s properties” (L355-356) is misleading. Would it be possible for the model to effectively memorize the poses in association with properties, as the orientation correlates with the type of molecule? For example, $\epsilon_{LUMO}$, ZVPE, and $c_V$ are all invariant properties, so the orientation physically should not hold information about the property. Rather, it should be made clear that this is a dataset level artifact. The model is essentially learning biases in the dataset’s conventions, rather than some physically meaningful relation between orientation and molecular property. I believe this may be what the authors meant but clarification would be helpful.

I believe these weaknesses, particularly in terms of novelty, are somewhat substantial.

**Questions:**

In Table 1, were any other QM9 properties explored? It would be interesting to see if the performance varies per property (although I expect it would not).

For Figure 6, it would be helpful to label each plot in a larger font with the chemical property being plotted.

In the conclusion L438, the authors state “it is essential to evaluate equivariant models on a randomly oriented test set as a sanity check for true equivariance.” Would it not be more robust to test the equivariance error of the model? E.g. for some sample $x$ and a rotation $R$
$$
\epsilon(x, R) = \frac{\| f(R \cdot x) - D(R) \cdot f(x) \|}{\| f(x) \|}
$$
and then average over multiple rotations/the data distribution.

Could the authors elaborate on/connect their work to prior work on functional vs. distributional symmetry breaking? E.g. see [5]. Functional symmetry breaking is where the mapping between inputs and outputs is not fully equivariant. Approximately equivariant models have mostly been applied in this setting. The setting studied in this paper is distributional symmetry breaking, where a datapoint $x$ and its transform $Rx$ are not equally likely under the data distribution. This is also stated in [1]. Given that approximately equivariant models are built for the setting of functional symmetry breaking, it is not clear that they “might artificially inflate their performance by exploiting extrinsic canonicalization” (L082).
In general, I would like the authors to relate their work to the classifier test presented in [1] and explain the novelty of their work.

[1] Lawrence et. al, Detecting Symmetry-Breaking in Molecular Data Distributions, https://openreview.net/forum?id=yEvdOXW5iY#discussion

[2] David Lopez-Paz and Maxime Oquab. Revisiting classifier two-sample tests. In International Conference on Learning Representations, 2017. URL https://openreview.net/forum?
id=SJkXfE5xx.

[3] Kenny Chiu and Benjamin Bloem-Reddy. Non-parametric hypothesis tests for distributional group symmetry. In NeurIPS AI for Science Workshop, 2023.

[4] Ashkan Soleymani, Behrooz Tahmasebi, Stefanie Jegelka, and Patrick Jaillet. A robust kernel
statistical test of invariance: Detecting subtle asymmetries. In Yingzhen Li, Stephan Mandt,
Shipra Agrawal, and Emtiyaz Khan (eds.), Proceedings of The 28th International Conference
on Artificial Intelligence and Statistics, volume 258 of Proceedings of Machine Learning Research, pp. 4816–4824. PMLR, 03–05 May 2025. URL https://proceedings.mlr.press/v258/
soleymani25a.html.

[5] Rui Wang, Elyssa Hofgard, Robin Walters, and Tess Smidt. Discovering symmetry breaking in physical systems with relaxed group convolution. arXiv preprint arXiv:2310.02299, 2024c.

---

> ### Author Response · Authors · 2025-11-22
> **Answer to Reviewer YdZo**
>
> Thank you for your thorough and constructive review. We are happy to hear that you find our paper well-written and that you appreciate our visualizations and the interesting patterns they reveal. Thanks as well for the questions and criticism, which we address in the following.
>
> ### 1 Relating to the AI4Mat workshop paper by Lawrence et. al (2025)
>
> > The authors should clarify the novelty of their work relative to [1].
>
> Many thanks for bringing Ref. [1] to our attention. We sincerely did not know this prior work and really appreciate its contribution. We agree with the reviewer that [1] holds precedence in introducing the idea that canonicalization biases do exist in molecular datasets. After a very thorough reading and appraisal of [1], we came to the conclusion that the substance of the present manuscript differs very significantly from [1] and indeed offers meaningful complementary insights and tools.
>
> While the focus of Ref. [1] is the detection of orientation bias in datasets, our investigations go beyond the analysis in [1] by examining the distribution of orientations in molecular datasets: We summarize the distribution of orientations by pairwise distances, quantitatively identify the most common orientations and, as one of our main contributions, offer an easy-to-use, faithful (equal-area) and interpretable visualization technique to jointly visualize all orientations in a molecular dataset (see e.g. Fig. 1 or Fig. 5). In our opinion, these contributions offer a significant and novel addition to Ref. [1] for investigating not only *if* orientations in a dataset are canonicalized but also *how*.
>
> Similar to our classification test in Sec. 3.1, Ref. [1] presents a two-sample classifier test to detect whether a molecule stems from the original (canonicalized) dataset or has been randomly rotated. We now explicitly credit Ref. [1] for this in the revised manuscript in l. 046 and l. 238. However, we would also like to highlight one subtle but novel aspect in our two-sample classifier test relative to [1]: Our test not only studies whether a neural network is able to detect canonical orientations but also how robust this detection is by applying increasingly strong perturbations to the canonical atom positions (rotations up to a maximum angle $\alpha$ and Gaussian jitter with different noise levels $\delta$, cf. Fig. 3). For instance, a rather weak alignment of a single chemical bond is no longer detectable after adding sufficient noise (as discussed in l. 241f). As such, our test still provides some additional insights compared to the classifier test carried out in Ref. [1].
>
> Furthermore, our work goes beyond the analysis of Ref. [1] by demonstrating that molecular orientations are in fact correlated with molecular properties. This means that the canonicalization of orientations in molecular datasets can in principle be leveraged by neural networks as shown in Sec. 3.3 (see also answer 7 below).  Beyond the detection of orientation bias, this is an important and novel finding which, in our opinion, should be shared with the community to avoid that any spurious information contained in the molecular orientations is leveraged in machine learning pipelines.
>
> Lastly, we believe that the new OMol25 dataset, which combines some of the most widely used community datasets (see Fig. 7), will soon become one of the central benchmarks in the molecular machine learning community. In that sense, our analyses on QMugs and OMol25 are timely and valuable additions to the datasets addressed in Ref. [1].
>
> [1] Lawrence et. al, Detecting Symmetry-Breaking in Molecular Data Distributions, https://openreview.net/forum?id=yEvdOXW5iY#discussion (2025)
>
> ### 2 Additional related work
>
> > Additionally, in relation to section 3.1, two-sample classifier tests have already been presented in the literature, so I believe [2] should be cited. There also exists literature on non-parametric hypothesis tests for distributional group symmetry and statistical tests for invariance [3,4], so these works should be cited or discussed.
>
> Thanks for pointing out these related works. We have added a citation of the mentioned Ref. [2] in the context of the two-sample classifier test (l. 237) and relate our non-parametric $\mathrm{SO}(3)$-uniformity test (added during the rebuttal) to Ref. [3] and [4] in the revised manuscript (l. 364).

---

> > ### Author Response · Authors · 2025-11-22
> > **Answer to Reviewer YdZo (continued)**
> >
> > ### 3 Relevance and novelty of our visualizations
> >
> > > While the visualization of molecular principal components are quite aesthetically pleasing, I am also not sure of their novelty. It is reasonable to expect that a machine learning practitioner would perform PCA on their data or visualize certain molecules prior to training a model, but much of the paper seems to rely on these visualizations. For example, take the observation that "chemically similar molecules tend to have similar canonical orientations."
> >
> > "A striking characteristic of human memory is that pictures are remembered better than words" [2] and, we claim, certainly better than numbers. It is in this sense that we perceive the extensive use of visualizations, and the emphasis on visualization tools, as a particular strength of our manuscript.  While we would very much wish that, prior to training a model, every ML researcher would investigate their data for biases, our experience shows this is not the case. Even when a selection of 3D molecular geometries is studied, subtle biases as in the case of the OMol25 Biomolecules subset (Fig. 7i) may easily be overlooked.
> >
> > Regarding the novelty of our visualizations: it is important to point out that our visualizations showing all molecular orientations in a dataset are *not* typical PCA plots. Usually, 2D PCA plots show higher-dimensional data projected onto the two orthogonal axes ($w_1, w_2$) that explain most of the data's variance, that is, the reconstruction $w_1 (w_1 \cdot x) + w_2 (w_2 \cdot x)$ should be as close to the original $x$ for all data points as possible. The usual PCA plot then shows a 2D scatter plot of the PCA scores $w_1 \cdot x$ and $w_2 \cdot x$. Instead, our figures visualize the orientations of molecules by plotting the directions of the principal components themselves in a Mollweide projection, colored by PC1 $\sim$ blue, PC2 $\sim$ and PC3 $\sim$ magenta (cf. Fig. 1).
> >
> > While prior work, e.g. [3], also visualizes elements from the rotation group $\mathrm{SO}(3)$, to best of our knowledge, we are the first to visualize rotation matrices as a set of row vectors as described in Sec. 3.4. Compared to the visualization in [3], our method has two advantages: a) splitting up the PC plots and color-coding chemical properties as in Fig. 6 is straightforward and b) if one changes the global coordinate system (or equivalently rotates all molecular geometries in a dataset by the same rotation) the pattern in our projections will simply rotate across the Mollweide projection. For the visualization in [3] both the colors will change and the scattered points will transform in a non-trivial way.
> >
> > In contrast to our visualizations, the figures 6 and 14 in Ref. [1] display the distribution of *principal moments* of inertia (although the titles of these figures incorrectly say "Distribution of Principal Components"). Thus, the figures in Ref. [1] convey very different information from our plots. Crucially, the moments of inertia (or, in the context of PCA, the eigenvalues corresponding to the PCs) are *rotationally invariant*. Therefore, the histogram of principal moments of inertia does not characterize the orientations of molecules but rather "how spherical the molecules are".
> >
> > Indeed, our conclusions and main messages are strongly supported by our visualizations of molecular orientation distributions but in no place do our conclusions rely on the visualizations alone. For instance, the statement that "chemically similar molecules tend to have similar canonical orientations" is supported by the considerations in Sec. 3.3, in particular by the experiment reported in Tab. 1. Since our visualization technique is very straightforward to use and requires little computational resources (no GPU), it may for many researchers be the first tool in line to detect and characterize orientation bias in geometric datasets, before training a neural network classifier.
> >
> > Altogether, we believe that our visualizations of all molecular orientations in the selected molecular benchmarks are novel and strongly emphasize the main messages of our paper.
> >
> > [2] C.L. Grady, A.R. McIntosh, M.N. Rajah, & F.I.M. Craik, Neural correlates of the episodic encoding of pictures and words, Proc. Natl. Acad. Sci. U.S.A. 95 (5) 2703-2708, https://doi.org/10.1073/pnas.95.5.2703 (1998).
> > [3] Murphy, Kieran A., et al. "Implicit-PDF: Non-Parametric Representation of Probability Distributions on the Rotation Manifold." International Conference on Machine Learning. PMLR, 2021.

---

> > > ### Author Response · Authors · 2025-11-22
> > > **Answer to Reviewer YdZo (continued)**
> > >
> > > ### 4 Data generation and canonicalization for QM9
> > >
> > > > For QM9, this most likely follows directly from the way conformers are generated (via CORINA from SMILES strings). Since CORINA uses deterministic geometry generation rules, this would be expected (the authors also state this). Therefore, I am not sure of the novelty of this finding/if it needs to be presented in the main part of the paper.
> > >
> > > As pointed out by the reviewer, we intend to give a putative explanation for the alignment of molecular poses in the QM9 dataset based on the data generating process (l. 208f). In particular, we point out the alignment of the chemical bond adjacent to the origin which supports the hypothesis that the alignment is caused by the geometric heuristics in the Corina algorithm (see also answer 2 to Reviewer wjTX). Indeed, we believe that cheminformatitions are well aware of the fact that computational chemistry codes used in the underlying data generating processes usually do not generate molecular geometries in random orientation (l. 014). However, at the same time, it can also be a real pitfall for machine learning practitioners and should in our opinion be stated explicitly. As an example, we point out two QM9 based charge density prediction datasets which evaluate the electron density on regular Cartesian (axis-aligned) grids that are not spherically symmetric and therefore biased towards the canonicalized orientations (see answer 6 to Reviewer pzKZ for more information).
> > >
> > > ### 5 Data generation and canonicalization in QMUGs and OMol25
> > >
> > > > For the other datasets (QMugs and OMol25) the authors should add further information as to why they think it has an orientational bias (L155).
> > >
> > > Regarding an explanation for the origin of the orientational bias in the QMugs and OMol25 datasets, please take a look at answer 2 to Reviewer wjTX.

---

> > > > ### Author Response · Authors · 2025-11-22
> > > > **Answer to Reviewer YdZo (continued)**
> > > >
> > > > ### 6 Non-equivariant models on equivariant tasks?
> > > >
> > > > > From my understanding, equivariant models generally perform best for molecular property regression on many of these datasets (outperforming non-equivariant models e.g. https://benchmarks.rowansci.com/). If there is useful information contained in the molecular orientations, shouldn’t we expect the non-invariant models to do better?
> > > >
> > > > We agree that molecular ML tasks are a very good fit for equivariant architectures. Indeed, most rotationally-equivariant approaches have been developed in the context of molecular ML and on many benchmarks equivariant architectures are the state-of-the-art. However, we also notice a recent but growing trend to use non-equivariant architectures for molecular tasks, probably kicked off by very prominent examples such as Alphafold3 (as mentioned in l. 043). Other recent examples are mentioned in l. 044 of the revised manuscript.
> > > >
> > > > The motivation for using non-equivariant architectures on equivariant molecular ML tasks is two-fold: a) constraining models to exact equivariance complicates the architectural design, and specialized layers are computationally demanding (l. 40). And b) models with built-in equivariance "can be difficult to optimize and require careful hyperparameter tuning to train successfully" [4].
> > > >
> > > > Some of the recently proposed architectures do not use explicit data augmentation, making them susceptible to orientation bias in molecular datasets:
> > > >
> > > > 1) The authors of Ref. [4] relax the equivariance constraint in the intermediate layers of networks during training by introducing an additional non-equivariant term that is progressively constrained until one arrives at a fully equivariant solution. Therefore, orientation bias in the dataset could have an undesired effect on the optimization trajectory of such networks. Since the models in Ref. [4] are trained on the MD17 dataset, to complement the analysis on rMD17 performed in Ref. [1], we have added visualizations of the 10 different rMD17 datasets to our appendix (see Fig. 10). Indeed, all rMD17 subsets are strongly canonicalized, although the distributions of orientations look qualitatively very different.
> > > > 2) The models trained with REMUL [5] and the MLIP presented in [6] learn rotational equivariance approximately by minimizing an additional loss. The latter is trained on OMol25 and GNNs trained with REMUL are benchmarked on the MD17 dataset. The canonicalization of both datasets may influence the training and evaluation of these models.
> > > > 3) Motivated by the "bitter lesson" (Sutton, 2019), the conformer generation model presented in Ref. [7] is based on an efficient and scalable diffusion model that operates directly on 3D atom positions without enforcing rotational equivariance. The authors conduct experiments on QM9 and the GEOM dataset (strongly aligned, cf. our subfigure Fig. 7a). Notably, in Sec. 5.4 of Ref. [7] the authors stumble across the fact that randomly rotating their training set prior to training negatively impacts their performance and hypothesize that the reason may be that "the DFT simulations used to generate the data might be implicitly encoding a canonical coordinate system, which affects generalization if broken." (p. 8).
> > > >
> > > > We have added these points to our related work (Sec. 2.1) and are convinced that the authors of these and similar (future) works will find our results interesting and could very much profit from the methods presented in our manuscript.
> > > >
> > > > [4] Pertigkiozoglou, Stefanos, et al. "Improving equivariant model training via constraint relaxation." Advances in Neural Information Processing Systems 37 (2024): 83497-83520.
> > > > [5] Elhag, Ahmed A., et al. "Relaxed Equivariance via Multitask Learning." arXiv preprint arXiv:2410.17878 (2024).
> > > > [6] Elhag, Ahmed A., et al. "Learning Inter-Atomic Potentials without Explicit Equivariance." arXiv preprint arXiv:2510.00027 (2025).
> > > > [7] Wang, Yuyang, et al. "Swallowing the Bitter Pill: Simplified Scalable Conformer Generation." International Conference on Machine Learning. PMLR, 2024.

---

> > > > > ### Author Response · Authors · 2025-11-22
> > > > > **Answer to Reviewer YdZo (continued)**
> > > > >
> > > > > ### 7 Studying the empirical effect of orientation bias on NNs
> > > > >
> > > > > > To strengthen the paper, it would be good to have more experiments comparing models on molecular property prediction.
> > > > >
> > > > > We agree with the reviewer that, beyond our experiments reported in Tab. 1, 3 and 4, it is an interesting question which type of architectures and on which tasks (e.g. prediction of scalar quantities vs. tensorial quantities) are how strongly affected by orientational bias. To that end, we have implemented an adaptation of the Graphormer architecture [8] to study the effect of orientation bias on a transformer model commonly used on geometric data. By default, the model only uses rotationally-invariant radial embeddings in the attention mechanism. We have modified the architecture to further ingest a simple angular embedding of relative distance vectors $\frac{x_i - x_j}{\| x_i - x_j \|}$. Importantly, this naive radial embedding breaks the exact invariance of the architecture. We have trained and evaluated our invariant and non-invariant Graphormer adaptations on molecular property prediction with and without canonicalized molecular orientations. We find that the non-invariant models trained on the canonical data indeed strongly rely on the alignment of orientations. Further, we could demonstrate that in some cases the non-invariant models leverage the orientation bias in the dataset to artificially inflate their performance compared to models trained with randomized orientations (see App. E.5 for details).
> > > > >
> > > > > [8] Ying, Chengxuan, et al. "Do transformers really perform badly for graph representation?." Advances in neural information processing systems 34 (2021): 28877-28888.
> > > > >
> > > > > ### 8 Can orientations "hold information" about chemical properties?
> > > > >
> > > > > > The test in Section 3.3 is interesting, but the claim it is "empirical proof that canonical orientation alone holds information about a molecule's properties" (L355-356) is misleading. Would it be possible for the model to effectively memorize the poses in association with properties, as the orientation correlates with the type of molecule?
> > > > >
> > > > > We fully agree that scalar (invariant) quantities do not change as the orientation of a molecule is changed. Still, if molecules with similar chemical properties are oriented similarly (as an artifact of the data generating process), a neural network can be trained to map the orientation to those chemical properties (as shown in Sec. 3.3). The learned pattern is of course not a physical law but a result of the pose bias in molecular datasets. In that sense, the canonical orientations alone can be used to infer molecular properties (although, in isolation, the orientation of a single molecule does not hold any chemically relevant information). We have clarified this in l.415 of the revised manuscript.
> > > > >
> > > > > ### 9 Prediction of other QM9 properties from the molecule's orientation alone
> > > > >
> > > > > > In Table 1, were any other QM9 properties explored? It would be interesting to see if the performance varies per property (although I expect it would not).
> > > > >
> > > > > Indeed, we have now trained one separate MLP model to regress each (scalar) molecular property in the QM9 and QMugs dataset based solely on the principal components of the molecular geometries. The results are summarized in Tab. 3 and 4. Please take a look at answer 4 to Reviewer wjTX for more details.
> > > > >
> > > > > ### 10 Randomly rotated test set
> > > > >
> > > > > > In the conclusion L438, the authors state "it is essential to evaluate equivariant models on a randomly oriented test set as a sanity check for true equivariance." Would it not be more robust to test the equivariance error of the model?
> > > > >
> > > > > Of course testing the equivariance error of a model is sufficient to guarantee genuine equivariance and should be the method of choice (we now additionally mention it in l. 522). Still, as a very easy-to-implement sanity check we advertise using a randomly rotated test set for equivariant models (trained without augmentation) by default.

---

> > > > > > ### Author Response · Authors · 2025-11-22
> > > > > > **Answer to Reviewer YdZo (continued)**
> > > > > >
> > > > > > ### 11 Connection to functional vs. distributional symmetry breaking
> > > > > >
> > > > > > > Could the authors elaborate on/connect their work to prior work on functional vs. distributional symmetry breaking?
> > > > > >
> > > > > > As mentioned by the reviewer, functional symmetry breaking [9] is often used if the symmetry in the data is broken. The beam axis of particle colliders offers a good example. The underlying dynamics are fully Lorentz-equivariant but the rotational symmetry in the data is broken by the preferred direction of the beam axis. However, there are also several examples of models with relaxed/broken symmetry (or simply non-equivariant models) applied to problems which entail the full symmetry (see answer 6 above). Our work is concerned with the analysis of orientational bias which can be seen as an instance of distributional symmetry breaking [9] w.r.t. $\mathrm{SO}(3)$. Distributional symmetry breaking only affects models which are not (fully) equivariant, since equivariant models are agnostic to global orientations of the input by design (cf. l. 037f). We related our work to these concepts in l. 130f of the revised manuscript.
> > > > > >
> > > > > > [9] Wang, Rui, et al. "Discovering Symmetry Breaking in Physical Systems with Relaxed Group Convolution." International Conference on Machine Learning. PMLR, 2024.
> > > > > >
> > > > > > We hope that our answer adequately addresses your major concern regarding the positioning of our work relative to [1] and hope you will be satisfied with the clarifications and improvements made. We also hope that we were able to address your questions and look forward to discussing further.

---

> ### Comment · Reviewer_YdZo · 2025-11-27
> **Thank you for the detailed responses**
>
> Thank you to the authors for the detailed rebuttal and the additional experiments added to the paper/for answering my questions. I do agree with the authors that their visualizations go beyond those in Lawrence et. al, and I see their utility for practitioners. I’m updating the presentation score to 3, as I do think this work did a very clear job of presenting the problem and visualizations. Unfortunately, I still feel that the paper is lacking in novelty, as Lawrence et al. essentially introduces the idea that these molecular datasets have distributional symmetry breaking/the classifier test (which is a large part of this paper). The new experiments are quite interesting, but I don’t feel that they add enough information to make concrete conclusions.
>
> It’s hard to make recommendations for practitioners without having more experiments relating to model selection. For example, for the QM9 properties shown in Table 1, there is a clear signal that the model is learning from the poses (it could be good to add percentage gain as in Table 3 and 4). However, for all of the QM9 properties in Table 3, it is not as clear (e.g. the percent improvement is ~1% for certain properties). It could be interesting to look into why the information varies per property (e.g. perhaps using the physical definitions). Additionally, the properties that the Graphormer is trained on are limited for Table 5. For QM9, ZVPE is the property that has the strongest signal from Table 3 (the prediction from poses), so it makes sense that the non-invariant model would use this information (as the authors state). Would it be useful to look into the other properties that do not have as high of a signal? I do think these experiments hold promise, but I think they are somewhat preliminary, and thus I am maintaining my score.

---

> > ### Author Response · Authors · 2025-11-30
> >
> > Many thanks for acknowledging our rebuttal. We are glad to hear that we could answer your questions. Thank you also for pointing out the utility and novelty of our visualizations and for highlighting our clear presentation. Let us again compactly summarize the respective contributions of Lawrence et al. ("Detecting Symmetry-Breaking in Molecular Data Distributions") and this work:
> > | Contribution         | Lawrence et al. | Our work | Where in our work  |
> > |------------|:-------:|:-------:|:----------------:|
> > | Detection of orientation bias with two-sample classifier  | ✓  | ✓  | Sec. 3.1 + Fig. 3  |
> > | Detection of orientation bias with MMD (as baseline)  | ✓       | ✗       | - |
> > | Visualization of all molecular orientations via Mollweide projections  | ✗       | ✓       | Sec. 3.4 + Fig. 1, 5, 7 |
> > | Analysis of orientation distribution via pairwise distances (identification of most common orientations) + Kolmogorov-Smirnov test for SO(3)-uniformity | ✗       | ✓        | Sec. 3.2 + Fig. 4 + App. F.2 |
> > | Revealing correlation between molecular properties and canonical orientations + experiments on empirical effect | ✗       | ✓        | Sec. 3.3 + Tab. 1, 3, 4 +  Fig. 6 + App. E.5  |
> > | Discussion of implications of orientation bias (including prior work that may be affected)   | ✗       | ✓       | l. 078f + Sec. 2.1    |
> >
> > Regarding concrete conclusions:  Our experiments (Tab. 1, 3, 4, 5) clearly demonstrate that many molecular properties in standard benchmarks are correlated with the canonical orientation of the molecules. Therefore, in our opinion, the practical recommendation is really clear: if you are training a non-equivariant model, please randomly rotate your dataset to avoid that the model adopts the orientational bias in the data, i.e. learns unphysical patterns that won’t generalize. More subtle implications are discussed in Sec. 2.1. These conclusions are valid, despite the fact that the correlation between molecular properties and molecular orientations varies in strength. Practitioners should be aware of the existence of orientation bias and its implications to be able to critically examine their own molecular ML pipeline.

---

### Official Review · Reviewer_wjTX · 2025-11-01

**Soundness:** 3
**Presentation:** 4
**Contribution:** 3
**Rating:** 8
**Confidence:** 4

**Summary:**

The paper demonstrates that the molecules in the most commonly used molecular benchmark datasets are not in random orientations. Worse, the orientation implicitly encodes information about some of the properties that algorithms are supposed to be predicting.

**Strengths:**

The paper makes an important observation about the most popular molecular benchmarks and demonstrates it quite convincingly. It is important to convey this message to the community because it might be biasing our understanding of the relative strengths of different architectures. It is remarkable that this issue persists across multiple datasets.

**Weaknesses:**

- The degree of deviation from uniformity of the orientations could be summarized in a single number, I don't see that in the paper.
- It would be very interesting to see a putative explanation for why the way the datasets were generated lead to bias in orientation.
- It is remarkable that a 3-layer message passing architecture is sufficient to decode the hidden signal in the orientations. I would be interested to see what the minimal architecture is that can do this. Relatedly, it would be interesting to see a how well this strategy works on all the target properties, not just the few cherry-picked ones reported in Table 1.
- Section 3.2 suggests that to some extent the molecules are aligned by their principle components. A natural question then might be whether a very simple model trained on just the principle eigenvectors could do the same as the message passing network.
- Of course it would be great to see how each of the models mentioned in the paper (both fully equivariant and non-equivariant) actually do on a randomly rotated version of these benchmarks, but I understand that running all the models is somewhat outside the scope of this paper.

**Questions:**

see "weaknesses" for how the paper could be improved

---

> ### Author Response · Authors · 2025-11-22
> **Answer to Reviewer wjTX**
>
> Thank you for your detailed and constructive review. We are happy to hear that you find our investigations important and our methods convincing. Thanks as well for the questions and criticism, which we address in the following.
>
> ### 1 Measuring the non-uniformity of orientations in a single number
>
> > The degree of deviation from uniformity of the orientations could be summarized in a single number, I don't see that in the paper.
>
> During the rebuttal we have devised an additional non-parametric statistical test to measure $\mathrm{SO}(3)$-uniformity in a single number. For that, we estimate the Kullback–Leibler (KL) divergence between the empirical distribution of orientations (as given by the principal components, cf. Sec. 3.2) and the uniform distribution of rotations via a version of the classical Kozachenko-Leonenko estimator [1] adapted to the curved domain of $\mathrm{SO}(3)$.
>
> Our estimated Kullback-Leibler divergences align very well with the visual impressions from the Mollweide plots:
> Ordering the OMol25 subsets by this non-uniformity measure yields the identical order that we had determined solely based on our visualizations, except for a single transposition of two consecutive subsets.
> In a revised version of Fig. 7 we use this updated order and include annotations with the estimated KL divergences.
>
> Please take a look at l. 353f and App. C for further details.
>
> [1] Singh, Shashank, and Barnabás Póczos. "Analysis of k-nearest neighbor distances with application to entropy estimation." arXiv preprint arXiv:1603.08578 (2016).
>
> ### 2 Explaining the orientation bias
>
> >It would be very interesting to see a putative explanation for why the way the datasets were generated lead to bias in orientation.
>
> For QM9 we intend to give a putative explanation in l. 208f. We suspect that the orientation bias may be explained from the observation that (almost) all molecular geometries in Figure 2 share a common bond (pointing along the Cartesian y-axis up from the origin). The authors of the QM9 paper report that the 3D molecular geometries were generated using the commercial cheminformatics tool Corina (Version 3.491 2013), which uses bond lengths, bond angles and other heuristics to produce atom positions form SMILES strings (as described in [2]). It seems reasonable that the algorithm would start the construction of Cartesian (xyz) coordinates by placing one chemical bond adjacent to the origin in the direction of the y-axis. This step in the data generation is most likely the origin of the orientation alignment in QM9. We assume the reason that the alignment of this first bond is not exact is that the geometries generated by Corina were optimized further using Kohn-Sham DFT calculations. Unfortunately, we are not able to validate this hypothesis empirically since the Corina algorithm is closed-source. In the online manual (https://mn-am.com/wp-content/uploads/2023/09/corina_classic_manual.pdf) we could not find a precise description of the coordinate generation. Although the documentation mentions an option to orient generated 3D structures according to their principal moments of inertia (p. 5), our analysis suggests that this option was not used in the generation of QM9.
>
> The molecules in QMugs were molecules extracted from the ChEMBL database. The original paper states that the 3D molecular geometries are obtained from SMILES strings using the implementation of the Experimental-Torsion Knowledge Distance Geometry (ETKDG) [3] available in the open-source cheminformatics library RDKit (http://www.rdkit.org). We have refrained from a more detailed investigation of the ETKDG implementation, as it requires a deep dive into the C++ code of RDKit.
>
> Since OMol25 is a collection of several datasets, it is even more difficult to pin the existing orientation bias to exact steps in the data generating process. However, for certain subsets the original OMol25 paper mentions how 3D geometries were generated: Some structures of the electrolytes subset were take from MD trajectories (p. 40), some were generated from SMILES strings using the Software for Chemical Interaction Networks Molassembler package (p. 47) and some initial structures for complexes of electrolytes were generated using the Architector package (p. 48).
>
> Side note: In particular, for the subsets of the rMD17 molecular dynamics dataset (Fig. 10 of the revised manuscript), it is surprising that the orientation distributions are qualitatively so different.
>
> [2] Jens Sadowski and Johann Gasteiger. From atoms and bonds to three-dimensional atomic coordinates: automatic model builders. Chemical Reviews, 93(7):2567–2581, 1993.
> [3] Riniker, S. & Landrum, G. A. Better informed distance geometry: Using what we know to improve conformation generation. J. Chem. Inf. Model. 55, 2562–2574 (2015).

---

> > ### Author Response · Authors · 2025-11-22
> > **Answer to Reviewer wjTX (continued)**
> >
> > ### 3 Shrinking the message passing classifier
> >
> > > It is remarkable that a 3-layer message passing architecture is sufficient to decode the hidden signal in the orientations. I would be interested to see what the minimal architecture is that can do this.
> >
> > Indeed, it is an interesting and relevant question to study the effect that shrinking the network size has on the classifier's accuracy. Reducing the number of trainable parameters lowers the train times and hardware requirements for probing a dataset for orientation bias. To that end, we have conducted an ablation study on QM9 by training a 2-layer and 1-layer message passing model. Both models still accurately discern the canonical dataset from a randomly rotated one (see Fig. 11 of the revised manuscript). Further, we have drastically reduced the number of learnable parameters on the 1-layer network from 642,000 to a "tiny" version with 25,000, to a "nano" version with 2350, and to a "pico" version with only 238 trainable parameters. Very surprisingly, even the smallest versions only sacrifice a bit of accuracy but still adequately detect the orientation biases in all three datasets. This result seems to illustrate the remarkable inductive bias in the geometric message passing network and suggests that in practice the classifier test in Sec. 3.1 may be performed with a much smaller network.
> >
> > ### 4 Property prediction from a molecule's orientation alone
> >
> > > Relatedly, it would be interesting to see a how well this strategy works on all the target properties, not just the few cherry-picked ones reported in Table 1.
> >
> > We have now trained one separate MLP model to regress each (scalar) molecular property in the QM9 and QMugs dataset based solely on the principal components of the molecular geometries. In Tab. 3 and 4 of the revised manuscript we report the mean squared error (MSE) that the MLP achieves when trained on the canonical datasets and on randomly rotated versions of them. We compare the MSE against the optimal test performance achievable for uninformative input features (as in Tab. 1). For almost all properties we find that the MLPs trained on the canonical datasets achieve a significantly lower MSE than the mean of the test set. For several properties the relative improvements are comparable to those of the selected properties shown in Tab. 1.
> >
> > ### 5 Training an MLP Classifier to discern orientations directly
> >
> > > I think Section 3.2 suggests that to some extent the molecules are aligned by their principal components. A natural question then might be whether a very simple model trained on just the principal eigenvectors could do the same as the message passing network.
> >
> > We have conducted exactly this ablation study and have trained vanilla MLPs ingesting only the molecule's orientation (in the form of the principal components) to predict whether the PCs come from a previously rotated molecular geometry or not. Conceptually, this test is similar to our non-parametric uniformity test (explained above). Indeed, the MLP classifier can still distinguish well between samples from the two distributions. However, since the principal components arguably contain fewer cues about a molecule's orientation than the full geometry, it seems plausible that the classifier is overall less accurate than the message passing network. For instance, in some cases the canoncial orientation may also be detectable from the orientation of local substructure in the molecule -- information which is not contained in the PCs.
> >
> > ### 6 Systematic comparison of molecular ML approaches on standard and rotated versions of benchmarks
> >
> > > Of course it would be great to see how each of the models mentioned in the paper (both fully equivariant and non-equivariant) actually do on a randomly rotated version of these benchmarks, but I understand that running all the models is somewhat outside the scope of this paper.
> >
> > This is a very interesting but unfortunately very time-intensive endeavor. While adapting the training codes and training the models in different dataset settings lies beyond the scope of our work, we view it as an interesting direction for future research (and possibly, a shared community effort). As a first step in this direction, we have trained a Graphormer model in the different settings (see answer 7 to Reviewer YdZo for more information).
> >
> > We hope we were able to address your questions and look forward to discussing further.

---

> > > ### Comment · Reviewer_wjTX · 2025-11-25
> > >
> > > Thank you for these clarifications, maybe some of these points could be mentioned in the discussion or conclusion sections of the paper. Please cite"Lawrence et. al, Detecting Symmetry-Breaking in Molecular Data Distributions" as mentioned by one of the other reviewers. I do not wish to change my review score.

---

> > > > ### Author Response · Authors · 2025-11-30
> > > >
> > > > Many thanks for your response. We are happy to hear that we could clarify your questions. In the revised version we cite the mentioned paper by Lawrence et al. We will additionally mention the ablations and some of their interesting results in the discussion/conclusion in the next revision.

---

### Author Response · Authors · 2025-12-02
**Summary of Revisions**

Once again, we would like to thank all reviewers for their time and thoughtful feedback. In summary, in response to all of the helpful comments, we have
* extended our experiment that reveals correlation between molecular properties and canonical orientations to all molecular properties in the selected benchmarks (Tab. 3, 4).
* conducted an additional experiment that studies the empirical effect of orientation bias on a typical geometric transformer architecture (App. E.5).
* expanded the discussion of the implications of orientation bias by highlighting prior work that may be affected by orientation bias (Sec. 2.1). For that, we demonstrate orientation bias in two additional molecular dynamics datasets (Fig. 9, 10).
* related our work to the concepts of functional/distributional symmetry breaking (l. 130f) and to the prior work by Lawrence et al. (l. 126f).
* conducted two additional ablations on the detection of orientation bias with neural classifiers: (i) the size of the geometric message passing classifier can be drastically reduced without sacrificing much accuracy (App. E.2) and (ii) an MLP that receives only principal components as input can still detect orientation bias, but less effectively than a message passing classifier that uses the full 3D molecular geometry (App. E.4).
* devised two additional non-parametric statistical tests to quantify the degree of orientation bias: (i) we estimate the KL divergence to the uniform distribution on SO(3) (App. C) and (ii) conduct a Kolmogorov-Smirnov test to compare the distribution of pairwise distances between orientations to the theoretical reference (App. F.2).

We are sorry we could not discuss more with you, the reviewers, for reasons beyond all of our control; but we truly value your input and have done our best to accommodate it. We much hope you like the outcome.

---

### Meta-Review · Area_Chair_YMeh · 2026-01-17

**Summary:**

Reviewer 1

1.Numerical results of non-uniformity of orientations.
2.Explainiation of bias.
3.Systematic comparison of molecular ML approaches on standard and rotated versions of benchmarks.
4.Expansion of QM9 results.

Reviewer 2

1. Lack of novelty due to a concurrent/prior work (Lawrence et al., AI4Mat 2025)
2. PCA visualization is standard practice and that the observation of chemically similar molecules sharing similar orientations is merely an expected artifact of deterministic conformer generation tools (e.g., CORINA).
3. Lack of experiments comparing equivariant vs. non-invariant models to support the claim that orientation bias "artificially inflates" performance.

Reviewer 3

1.Code for reproducibility
2.Randomized scattering in Mollweide plots

Reviewer 4

1.Confusion between "Pose Bias" and Physical Conformation
2.Empirical Verification on Trained Models. Should test trained models, such as EGNNs, to see if they are empirically susceptible to variations in random rotations.
3.Relevance to Equivariant Models

**Reviewer Concerns:**

Based on the review summary, the rebuttal successfully addressed several technical concerns, including the clarification that "pose bias" stems from deterministic generation tools (e.g., CORINA) rather than physical conformational preference (Reviewer 2, Reviewer 4), the justification of the study's relevance to relaxed-equivariant architectures (Reviewer 4), and the request for expanded QM9 property analysis (Reviewer 1). While the defense of the visualization techniques led to an improved presentation score (Reviewer 2), the critical concern regarding the lack of novelty relative to Lawrence et al. remains outstanding (Reviewer 2). Furthermore, the new experiments demonstrating how non-invariant models exploit orientation bias were ultimately deemed "preliminary" and insufficient to robustly prove the extent of performance inflation (Reviewer 2, Reviewer 4).

**Reviewer Scores:**

I believe that all reviewers would likely maintain their original scores except Reviewer F9xk. It is likely that Reviewer F9xk would improve the rating from 4 to 6.

---

### Decision · Program_Chairs · 2026-01-26

Accept (Poster)